# CAN-CAP: CALIBRATION-FREE AND NOISE-RESILIENT HUMAN MOTION CAPTURE VIA LIDAR-CAMERA INTEGRATION

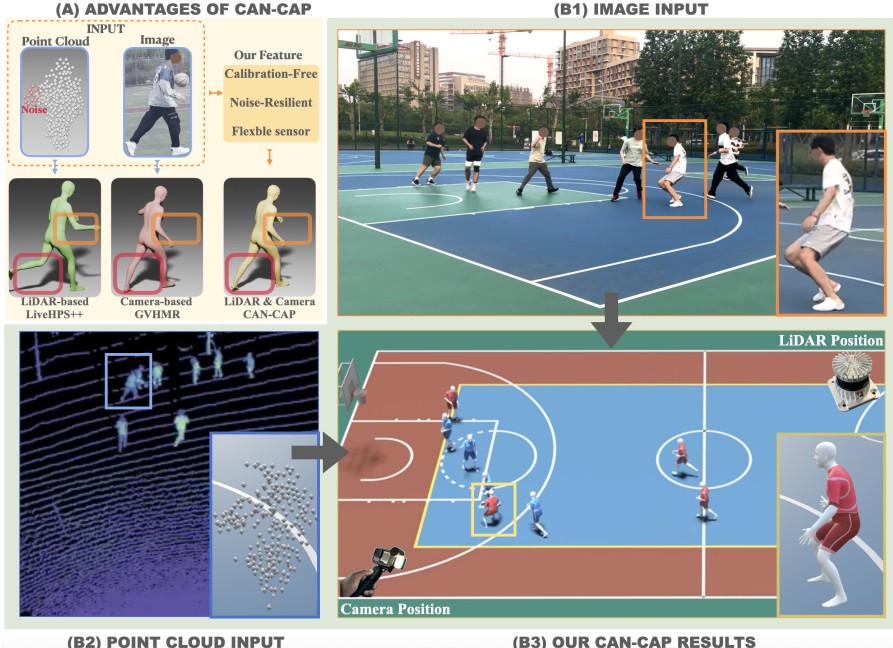

Figure 1. We propose (A) **Can-Cap**, a multi-modal motion capture framework that leverages the rich texture information from cameras together with the crucial depth cues from LiDARs. More importantly, Can-Cap addresses key limitations of prior multi-modal approaches by eliminating calibration dependencies, enhancing robustness to noise, and supporting flexible sensor combinations, thereby making it highly practical for real-world applications. (B) Application on a basketball example to illustrate the versatility of Can-Cap. The results (B3) confirm that Can-Cap successfully fuses heterogeneous inputs from a camera (B1) and a LiDAR sensor (B2), enabling accurate 3D human pose estimation in challenging dynamic scenarios without requiring sensor calibration. **Please refer to the appendix video for additional visualizations and real-world application results.**

## ABSTRACT

We propose **Can-Cap**, a **Ca**libration-Free and **N**oise-Resilient 3D human motion **Cap**ture framework that integrates multi-modal data from LiDAR and camera. While multi-modal sensors provide richer information than single-modal sensors, most existing approaches rely on pre-calibration for cross-sensor alignment, which propagates errors, especially when sensors have varying or dynamically changing perspectives. This reliance also requires fixed sensor placement with highly overlapping views, limiting flexibility and diminishing the benefits of diverse viewpoints for handling occlusions. Furthermore, prior methods often degrade under substantial noise or partial sensor failures, conditions common in real-world scenarios. To address these challenges, **Can-Cap** introduces a Unified Across-Sensor Motion Estimator that reconstructs local pose and shape in a human-centric space without calibrations between sensors, supporting flexible number of sensors, and a Noise-Resistant Trajectory Tracker that maintains

robustness under severe point cloud noise through iterative refinement. These calibration-free and noise-resilient features makes Can-Cap more practical in real-world deployment. Notable, operating in real time, **Can-Cap** achieves state-of-the-art results on Human-M3 and FreeMotion, as well as strong cross-domain performance on LiDARHuman and RELI11D. This combination of flexibility and robustness opens new opportunities for motion capture in real-world scenarios, e.g. sports analytics, field robotics, and large-scale immersive environments.

# 1 INTRODUCTION

Accurate and robust global 3D human motion capture (Xu et al., 2020; 2022; Jiang et al., 2023; Kaufmann et al., 2021; Yi et al., 2022; Charles et al., 2016; Belagiannis & Zisserman, 2017; Kim et al., 2019) is critical for applications in AR/VR, robotics, human-computer interaction, sports, and healthcare (Xu et al., 2018; Xin et al., 2024). Markerless motion capture methods typically use either cameras (Shin et al., 2024; Shen et al., 2024) or LiDAR (Ren et al., 2024b;a; Cai et al., 2023; Jang et al., 2023), But single-modality methods suffer from limited information: Monocular cameras lacks depth, while LiDAR lacks texture. Fusing both modalities provides complementary strengths, geometry and location information from LiDAR and appearance information from camera, as shown in Fig 1, enabling more precise human motion estimation of local pose and global translation.

Nevertheless, while multi-modal sensor methods provide richer information, they face increasing challenges as the number of sensors and modalities grows. On one hand, most multi-modal MoCap methods (Dou et al., 2017; Huang et al., 2018b; Qiu et al., 2019; Xue et al., 2025) rely on data alignment via calibration, which propagates errors, especially when sensors have varying or dynamically changing perspectives, and thus requires fixed sensor placement with highly overlapping views, limiting flexibility and reducing the benefit of diverse viewpoints for occlusion handling. Although the most recent SoTA FreeCap (Xue et al., 2025) eliminates manual calibration through on-the-fly estimation from matched keypoints, the involved calibration step can still accumulate errors, potentially leading to failures. On the other hand, prior methods degrade significantly under severe noise (Ren et al., 2024b). Although recent methods such as LiveHPS++ (Ren et al., 2024a) leverage LiDAR geometry to improve translation estimation, severe noise can cause normalization of point clouds to shift human points away from the valid data domain, resulting in substantial inaccuracies and unstable trajectories.

The above limitations in prior works motivate two central research questions: 1) *Can hybrid (LiDAR+Camera) motion capture be achieved in a flexible calibration-free manner?* This flexibility allows the system to adaptively adjust sensor deployment strategies to mitigate occlusions and address diverse task requirements, making motion capture more practical for real-world applications ; and 2) *How to make precise prediction of global human translation in highly noisy and cluttered environments?*

To address the challenges, as Figure 1 shows, we propose **Can-Cap**, a novel human motion capture approach that predict (1) human local poses through an innovative **calibration-free** multi-modal fusion strategy with a Unified Across-Sensor Motion Estimator (**UAME**), and (2) global trajectories with a **Noise-Resilient** Trajectory Tracker (**NTT**), achieve robustness and precision results even in noisy and variable viewing environments. The local poses and global trajectories together consist the final estimation of human motion. Specifically, the **UAME** that estimate the human local pose and shape in the *Human-Centric Space*. This design enables effective knowledge transfer across heterogeneous sensors through a Human-Centric Space Alignment mechanism and Fault-Tolerant Sensor Fusion, which dynamically unifies data from uncalibrated sensors with unknown poses, enabling easily deployment in complex indoor and large-scale outdoor environments. The tracker employs an iterative refinement mechanism that gradually expands the searching space, and even if the distribution of point clouds is greatly affected by extreme noise, it can achieve consistent and accurate global translation estimation. They together result in calibration-free global pose estimation that is scalable to variations in sensor modality, viewpoint and even quantity.

Beyond these core innovations, our approach makes a key architectural advances: A scalable sensor fusion architecture that encodes heterogeneous sensor inputs from different configurations into a unified human-centric representation space, enabling flexible deployment with variable sensor counts for expanded scene coverage and improved pose accuracy.

We evaluate *Can-Cap* against SoTA single-sensor and hybrid-sensor methods on the large-scale multi-person dataset Human-M3 (Fan et al., 2023a) and the multi-view dataset FreeMotion (Ren et al., 2024b). We further conduct cross-dataset evaluations on the challenging outdoor dataset LiDARHuman Fan et al. (2023a) and the fast-motion dataset RELI11D Yan et al. (2024), where viewpoint discrepancies relative to the training data are substantial. Extensive experiments demonstrate that our approach outperforms existing methods in robustness to noise and generalization capability, while maintaining calibration-free. Finally, we assess scalability by varying sensor counts under a unified parameter set, confirming the adaptability of our framework to diverse deployment scenarios. Our main contributions are summarized as follows:

1. We present *Can-Cap*, the first method to achieve calibration-free global human motion capture with a flexible number of multi-modal sensors (LiDAR and cameras).
2. We propose a Unified Across-Sensor Motion Estimator and a Noise-Resilient Trajectory Tracker that enable cross-modal alignment with uncalibrated, multi-modal, varying numbers of sensors, while remaining robust to substantial sensor noise.
3. We validate performance of *Can-Cap* extensively on multiple public benchmarks as well as through real-world deployments, demonstrating unprecedented flexibility and robustness for in-the-wild applications.

## 2 RELATED WORK

### 2.1 VISUAL SENSOR-BASED MOTION CAPTURE

Early motion capture methods that estimate high-quality human motions rely on wearable sensors, such as markers (OptiTrack; Raskar et al., 2007; Loper et al., 2014; Park & Hodgins, 2008; Song & Godøy, 2016; Xu et al., 2019), IMUs (Ren et al., 2023; Yi et al., 2021; Huang et al., 2018a), and ego cameras (EgoCap, 2016; Betancourt et al., 2017; Jiang & Grauman, 2017; Cao et al., 2017; Zheng et al., 2018) but are limited by issues such as sensor drift, discomfort for the wearer, and challenges in capturing certain movements, especially in dynamic or complex environments. The shift towards non-wearable motion capture has been significant with the use of cameras. Monocular-based methods (Pons-Moll et al., 2014; Kanazawa et al., 2018; Goel et al., 2023; Mehta et al., 2017b; Alldieck et al., 2017; Mehta et al., 2017a) provide convenient solutions but suffer from depth perception limitations and rely on single-modal representations that lack robustness. Multi-view-based methods (Dou et al., 2017; Huang et al., 2018b; Malleson et al., 2019) can capture comprehensive motion information but are difficult to set up and typically employ simple feature concatenation rather than principled cross-modal representation learning. Both of them are hard to recover the global human motions due to limited representation capacity. Recent efforts (Shin et al., 2024; Rajasegaran et al., 2022; Shen et al., 2024; Wang et al., 2024; Sun et al., 2023; Yuan et al., 2022; Ye et al., 2023) that use dynamic monocular views to estimate global motions still face challenges with translation accuracy, particularly in long video sequences, primarily due to insufficient temporal representation modeling. LiDAR has been instrumental in capturing global human motion in open and large-scale environments (Wu et al., 2024; Fan et al., 2023b). Early LiDAR-based methods like LiDARCap (Li et al., 2022) focus on local pose estimation using point-based representations, while LiDAR-HMR (Fan et al., 2023b) aims to reconstruct human meshes through geometric feature learning. However, these methods are limited by the lack of unified representation spaces that can effectively integrate geometric and semantic information. PointHPS (Cai et al., 2023) and LiveHPS (Ren et al., 2024b) have since advanced to predict full SMPL parameters, capitalizing on the depth information provided by LiDAR through improved point cloud. LiveHPS++ Ren et al. (2024a) further improves the robustness and smoothness of human motion capture via temporal consistency modeling. Despite these advancements, challenges for LiDAR-based methods still remain due to data sparsity and the absence of texture information, fundamentally stemming from the lack of cross-modal representation learning capabilities.

### 2.2 HYBRID SENSOR-BASED MOTION CAPTURE

Recognizing the limitations of single visual sensor-based approaches, research has increasingly focused on using multiple sensors (Xu et al., 2023). However, existing multi-modal approaches typically employ naive feature concatenation or late fusion strategies, lacking principled cross-modal

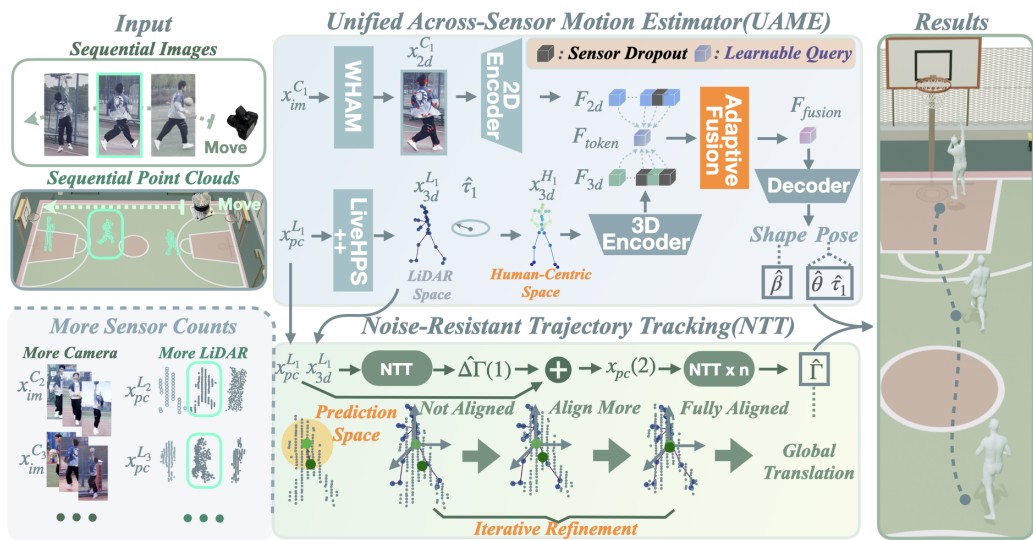

Figure 2: The pipeline of Can-Cap. With sequential point clouds from movable LiDARs and 2D key points from movable cameras as input, Can-Cap consists of two primary modules, Universal Motion Estimator for obtaining the human global orientation, pose, and shape, and Noise-resistant Trajectory Tracker for global trajectory estimation. We also show the details of Unified Across-Sensor Motion Estimator and Noise-resistant Trajectory Tracker. All components corresponding to our key contributions are highlighted in orange.

representation learning frameworks. Human-M3 (Fan et al., 2023a) integrates LiDARs with cameras to address occlusion challenges, and fully utilize advantages in different modal data through simple feature aggregation. However, these systems are often constrained by the need for precise calibration, necessitating fixed and pre-deployed setups, and more importantly, they lack unified cross-modal representation spaces that can effectively bridge the semantic gap between different modalities. A recent advance is FreeCap (Xue et al., 2025), which proposes a calibration-free method to match multi-persons in point clouds and images and uses the matching pairs to calculate a coarse calibration matrix. However, the network is heavyweight due to its lack of efficient representation learning mechanisms, and in cases where reference human points are few, the calibration matrix is inaccurate, degrading the performance of the fusion results. Additionally, it only supports one LiDAR within its framework, limiting its applicability and scalability. In contrast to these approaches that rely on explicit calibration or simple feature fusion, our method introduces a novel multi-modal representation learning framework that learns unified cross-modal representations through Human-Centric Space Alignment, eliminating the need for calibration estimation while supporting extendable LiDARs and cameras.

## 3 METHODOLOGY

Accurate 3D human motion capture from uncalibrated, heterogeneous sensors is challenging due to the inherent variability in sensor configurations and the susceptibility to noise in real-world environments. Existing methods often fail to address two critical issues: (1) *dependency on fixed sensor setups* that requires precise calibration and cannot adapt to dynamic sensor availability; (2) *sensitivity to sensor noise and occlusion* that leads to inaccurate global trajectory estimation.

To overcome these limitations, we propose **Can-Cap**, which tackles the problem through two technical innovations: first, a **Unified Across-Sensors Motion Estimator**(Section 3.1) that extracts consistent human pose and shape from flexible sensor combinations in a calibration-free manner; second, a **Noise-resistant Trajectory Tracker**(Section 3.2) that resists sensor noise through iterative refinement. Notably, our method supports input from varying combination of LiDARs and

cameras, offering high flexibility and adaptability in real-world deployments. An Overview of **Can-Cap** is shown in Figure. 2.

**Problem Statement and Preliminaries**     *Can-Cap* takes as input sequentially sampled point clouds $x_{\text{pc}}$ from LiDAR and normalized 2D keypoints $x_{2d}$ from images, requiring no calibration and supporting a flexible number of sensors, and predicts the full SMPL (Loper et al., 2015) parameters as the outputs, which represent the underlying human pose and shape. Although our input is time-series data, for the sake of convenience, we save time $t$ when writing symbols. First, we follow LiveHPS++ (Ren et al., 2024a) to use the farthest point sampling algorithm (FPS) to sample the point cloud as a fixed number $x_{pc}^{L_i} \in \mathbb{R}^{256 \times 3}$ from the i-th LiDAR, and follow WHAM (Shin et al., 2024) to normalize the 2D keypoints $x_{2d}^{C_i} \in \mathbb{R}^{17 \times 2}$ with its bounding box center and scale from the i-th Camera. Meanwhile, following (Xue et al., 2025), the ground truth of our algorithm is SMPL (Loper et al., 2015) parameters with shape $\beta \in \mathbb{R}^{10}$, body pose $\theta \in \mathbb{R}^{23 \times 3}$, global orientation $\tau \in \mathbb{R}^3$ and global translation $\Gamma \in \mathbb{R}^3$ while $\hat{\beta}$, $\hat{\theta}$, $\hat{\tau}$ and $\hat{\Gamma}$ are corresponding predictions of our framework. These parameters are used to describe the mesh of the human body with 6890 vertices and 24 joints. Finally, considering human matching among multi-sensors is a prerequisite for our calibration-free framework for multi-person scenarios, we adapted the matching mechanism of FreeCap (Xue et al., 2025) for real-time online inference. **More details are provided in the Section A of Appendix.**

### 3.1 UNIFIED ACROSS-SENSOR MOTION ESTIMATOR: LOCAL POSE RECOVERY

The core challenge in calibration-free motion capture is reconciling data from arbitrarily configured sensors. Our solution exploits the invariant properties of human kinematics: by estimating and aligning to the human centric space with *Unified Across-Sensor Motion Estimator (UAME)*, we create a consistent reference frame that automatically unifies inputs from diverse sensors, regardless of their number, type, or viewpoint. Details on architecture of the Unified Across-sensor Motion Estimator can be found in Appendix D.2.

As shown in Figure. 2, UAME takes as input uncalibrated sequential 3D point clouds $x_{\text{pc}}^{L_i}$ from multiple LiDARs $L_i$, and 2D keypoints $x_{2d}^{C_j}$ from multiple cameras $C_j$. It then estimates the local pose $\hat{\boldsymbol{\theta}}$ and human shape $\hat{\boldsymbol{\beta}}$ in a human-centric coordinate system that remains consistent across diverse sensor viewpoints and modalities. This is achieved through two key stages: 1) *Human-Centric Space Alignment*, and 2) *Fault-Tolerant Sensor Fusion*, aims to address sensor variability through an attention-based fusion module. By adaptively weighting the contributions from each sensor modality based on their reliability and relevance, it generates a fused feature that dynamically adjusts to the current sensor configuration, ensuring robust and accurate local pose estimation across diverse sensor configurations.

**Stage 1. Human-Centric Space Alignment**     Given sequential point clouds $\{x_{pc}^{L_i}\}_{i=1}^{N_L}$ from $N_L$ LiDARs, we first learn a unified representation space that aligns all heterogeneous sensor modalities to a Human-Centric coordinate system $H$. This alignment mechanism enables effective cross-modal knowledge transfer without explicit calibration. Human-Centric coordinate system is centered around humans, which is independent of the sensors. For each LiDAR $L_i$, a Gated Recurrent Unit(GRU) learns to predict rotation vector $\tau_i$ that transforms its 3D joint representations $x_{3d}^{L_i}$ encoded by 3D PointNet Encoder to the unified Human-Centric space $H_i$ from LiDAR $L_i$.

$$x_{3d}^{H_i} = R(\hat{\tau}_i)^{-1} x_{3d}^{L_i}, \tag{1}$$

where $R(\cdot)$ transforms rotation vector to rotation matrix. We then use loss $\mathcal{L}_{align}$ to optimize 3D representations in both LiDAR space $\mathbf{x}_{3d}^L = \{\mathbf{x}_{3d}^{L_1}, \cdots, \mathbf{x}_{3d}^{L_{N_L}}\}$ and human-centric spaces $\mathbf{x}_{3d}^H = \{\mathbf{x}_{3d}^{H_1}, \cdots, \mathbf{x}_{3d}^{H_{N_L}}\}$, together with 6-d representation of the global transformation matrix (Zhou et al., 2018) $\tau_i^{6d} = \{\tau_i^{6d}, \cdots, \tau_{N_L}^{6d}\}$ with

$$\mathcal{L}_{align}(\tau^{6d}, \mathbf{x}_{3d}^L, \mathbf{x}_{3d}^H) = \frac{1}{N_L} \sum_{i=1}^{N_L} (\|\tau_i^{6d} - \hat{\tau}_i^{6d}\|_2^2 + \|x_{3d}^{L_i} - \hat{x}_{3d}^{L_i}\|_2^2 + \|x_{3d}^{H_i} - \hat{x}_{3d}^{H_i}\|_2^2), \tag{2}$$

Besides, the 2D inputs are aligned by encoding the 2D keypoints $x_{2d}$ into a human-centric feature space via the 2D Encoder of WHAM (Shin et al., 2024).

**Stage 2. Fault-Tolerant Sensor Fusion.** This stage introduces an attention-based fusion module to mitigate sensor variability. By dynamically weighting the contributions from each sensor modality, it generates a fused feature that is invariant to the current sensor configuration, ensuring reliable local pose estimation.

We first learn modality-specific representations that capture the unique modal characteristics of each sensor type. The aligned 3D joint representations $x_{3d}^{H_i}$ are first processed through GRU to extract temporal motion features $F_{3d}$ that capture spatial-temporal dynamics, while sequential 2D keypoints $\{x_{2d}^{C_j}\}_{j=1}^{N_C}$ from $N_C$ Cameras are encoded via a GRU to produce visual motion features $F_{2d}$ that capture appearance-based motion patterns. Then, we introduce an adaptive fusion mechanism that learns to dynamically integrate heterogeneous sensor representations. A learnable query fusion token $F_{token}$ serves as a query to aggregate multi-modal features through cross-attention, enabling the model to adaptively weight different modalities based on their reliability and relevance:

$$F_{fusion} = \text{CrossAttn}(F_{token}, \text{Concat}(F_{3d}, F_{2d})) \tag{3}$$

where Concat denotes concatenation. The fused features are afterwards decoded through GRU to predict SMPL parameters $\hat{\beta}, \hat{\theta}$ with $\mathcal{L}_{pose}$:

$$\mathcal{L}_{pose}(\hat{\beta}, \hat{\theta}) = \alpha\|\beta - \hat{\beta}\|_2^2 + \|\theta - \hat{\theta}\|_2^2 \tag{4}$$

where $\hat{\beta}, \hat{\theta} = \text{GRU}(F_{fusion})$.

*Sensor Dropout* To enhance the model's robustness against incomplete inputs, we random dropout the features of sensors during training. Specifically, for each training sample, we randomly mask a certain percentage of feature tokens from both the LiDAR and camera modalities before they are fed into the fusion module. This strategy simulates various scenarios, from partial sensor data loss to complete sensor failure. This, in turn, enables dynamic adaptation to different sensor configurations at inference time without needing to retrain the model.

### 3.2 Noise-resistant Trajectory Tracker: Robust Global Localization

Besides a precise local motion, an accurate global trajectory is another critical component for coherent human motion capture. Thus, we then take as input the sequential point clouds $\mathbf{x}_{pc}$ from each LiDAR, and the corresponding 3D joints $\hat{\mathbf{x}}_{3d}^L$ (Section 3.1) to estimate the Global Trajectory $\hat{\Gamma}$.

Standard pipeline first normalized the point cloud so that its centroid coincides with the coordinate origin, then predicts a single offset to reach the human root. This procedure implicitly assumes that the centroid is close to the pelvis; however, noise or partial occlusion can shift the centroid far away, forcing the network to explore an abnormally large **prediction space** of candidate offsets. The enlarged prediction region biases the learned prior toward under-estimating large displacements and finally produces the systematic drift visualized in Figure. 3. To keep the prediction space bounded regardless of the initial perturbation, we adopt an iterative refinement strategy that gradually moves the centroid closer to the true root, reducing both offset magnitude and prediction variance at every step.

As a result, to obtain a robust and smooth global trajectory under sensor noise and environmental variations, we present a Noise-resistant Trajectory Tracker (NTT) with an iterative refinement mechanism. This approach progressively improves the trajectory through noise-resistant temporal modeling. For each step, we predict the offset $\Delta\hat{\Gamma}^n$ from the previous point clouds $x_{pc}^{n-1}$ in $n-1$ iteration and local motion $x_{3d}^L$, with

1) *Iterative Offset Prediction*: For each step $n$:

$$\Delta\hat{\Gamma}^n = E_O(x_{pc}^{n-1}, x_{3d}^L), \quad x_{pc}^n = x_{pc}^{n-1} + \Delta\hat{\Gamma}^n, \tag{5}$$

where $E_O$ is the offset predictor and we denote $x_{pc}^0 = x_{pc}$ and the center of original point cloud $\overline{x}_{pc}$.

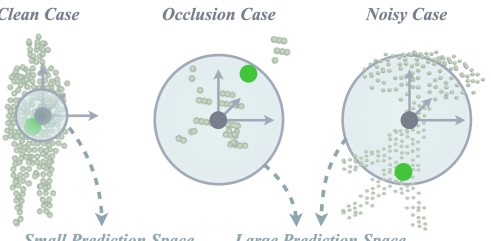

*Clean Case*  *Occlusion Case*  *Noisy Case*

*Small Prediction Space*  *Large Prediction Space*

Figure 3: Systematic drift arises when the root joint (green circle) is offset from the point cloud center $\overline{x}_{pc}$ (gray circle), especially under severe occlusion and noise .

2) *Constrained Optimization*: The final $\hat{\Gamma}$ of the human in LiDAR coordinates is then determined as $\hat{\Gamma} = \overline{x}_{pc} + \sum_{n=1}^{N} \Delta\hat{\Gamma}^n$ after $N$ iterations. We optimize the global trajectory representation with loss $\mathcal{L}_{tr}(\Gamma)$ using ground truth step offset and total trajectory, along with penalizing too large predictions,

$$\mathcal{L}_{tr}(\Gamma) = \|\hat{\Gamma} - \Gamma\|_2^2 + \lambda\frac{1}{N}\sum_{n=1}^{N}\mathcal{L}_{dtr}(\Delta\hat{\Gamma}^n), \tag{6}$$

where $\mathcal{L}_{dtr}(\Delta\hat{\Gamma}^n)$ is the penalty loss for preventing large step size with penalty $\delta$:

$$\mathcal{L}_{dtr}(\Delta\hat{\Gamma}^n) = \begin{cases} 0 & ,|\Delta\hat{\Gamma}^n| \le \delta \\ ||\Delta\hat{\Gamma}^n| - \delta| & ,|\Delta\hat{\Gamma}^n| > \delta. \end{cases} \tag{7}$$

**Remarks.** Unlike the coordinate system-independent parameters $(\hat{\beta}, \hat{\theta})$ estimated in the local representation (Section 3.1), the noise-resilient global trajectory $\hat{\Gamma}$ is defined relative to LiDAR coordinate systems. To this end, the Noise-resistant Trajectory Tracker(NTT) predicts the global translation for each LiDAR in a batched manner, depending on the number of available LiDARs.

**Real-Time Online Deployment.** For real-time online deployment, we additionally enhance temporal consistency and trajectory smoothness as detailed in Appendix A.

## 4 EXPERIMENTS

**Implementation Details** Our framework is implemented in PyTorch, trained with the AdamW optimizer and cosine learning rate decay. All experiments are conducted on NVIDIA RTX-3090 GPUs. Detailed hyperparameters and network architectures are provided in Appendix D.

**Baselines** involves **5** SOTA methods: 2 Camera based methods, WHAM (Shin et al., 2024) and GVHMR (Shen et al., 2024), 2 LiDAR-based methods, LiveHPS (Ren et al., 2024b) and LiveHPS++ (Ren et al., 2024a), and 1 Camera-LiDAR-based approach FreeCap (Xue et al., 2025).

**Datasets** We evaluate on large-scale multi-person Human-M3 (Fan et al., 2023a) and multi-view sensor FreeMotion (Ren et al., 2024b) datasets, following the experimental setting of FreeCap (Xue et al., 2025). We also evaluate cross-domain performance on outdoor LiDAR-camera dataset Li-DARHuman26M (Li et al., 2022) and indoor fast motion dataset RELI11D (Yan et al., 2024). Dataset details are provided in Appendix C.

**Metrics.** Following FreeCap (Xue et al., 2025), we evaluate our method using per-joint/vertex error in millimeters (**J/V Err(PS/PST)**), acceleration error in $m/s^2$ (**Accel Err**), per-global-joint rotation error in degrees (**Ang Err**), and scene-level unidirectional Chamfer distance in millimeters (**SUCD**).

### 4.1 COMPARISONS AND DISCUSSIONS

**In-domain Evaluations** Generally, as shown in Table 1, in both viewport settings, training view and novel camera viewpoints, our method demonstrates substantially superior performance to all baselines on both large-scale datasets FreeMotion and HumanM3. Under training view, thanks to our UAME module (Section. 3.1)'s capability of consistently fusing the complementary information from LiDAR and cameras, we significantly improve human motion capture performance compared to both camera-based methods, WHAM, and GVHMR, and Lidar-based methods, LiveHPS, and LiveHPS++. Meanwhile, we also greatly outperform the hybrid baseline, FreeCap, thanks to our human-centric alignment strategy, that effectively eliminate the need for explicit sensor calibration and thus greatly mitigate errors. Notice that when using only a single LiDAR, our method achieves comparable performance to LiDAR-only methods, LiveHPS and LiveHPS++, while the full multi-modal version yields substantial gains, underscores the effectiveness of our calibration-free architecture in effectively aligning and benefiting from diverse data in multiple multi-modal sensors.

Our LiDAR-only CanCap (**Ours**[†]) falls a bit short of LiveHPS++ on some FreeMotion metrics, but that's mostly because we skip the global joint velocity cues. LiveHPS++ uses a Velocity Predictor that smooths out tiny jitters in slow indoor movements, but it can get messy and drift when things speed up or change suddenly (see Figure 6 in the Appendix). We dropped that part to keep the

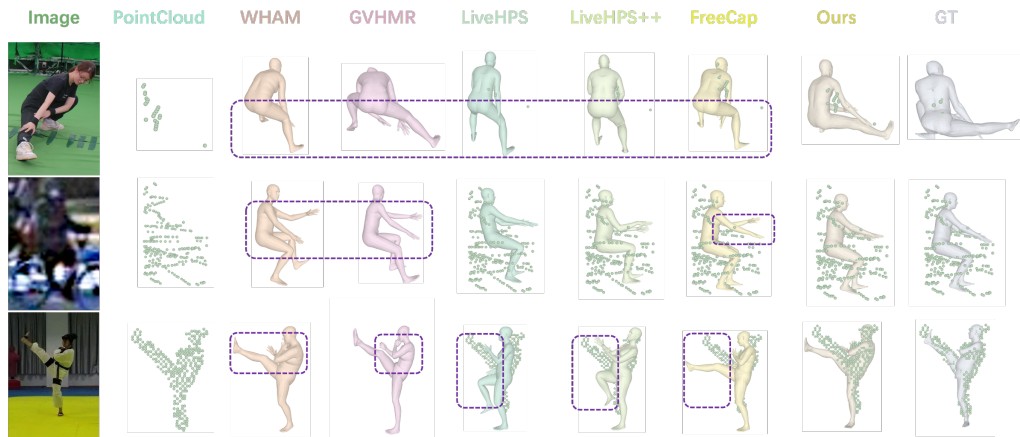

Figure 4: Qualitative comparisons showing Can-Cap's superior motion capture in challenging scenarios: (1) sparse-point leg stretching (2) Outdoor cycling (3) Fast-motion Taekwondo. We have highlighted the regions with significant differences using purple boxes.

| Methods | Venue&Year | Modality | View | Dataset FreeMotion (Ren et al., 2024b) | | | | | Dataset HumanM3 (Fan et al., 2023a) | | | | |
|---|---|---|---|---|---|---|---|---|---|---|---|---|---|
| | | | | J/V Err(PS)↓ | J/V Err(PST)↓ | Ang Err↓ | Accel Err↓ | SUCD↓ | J/V Err(PS)↓ | J/V Err(PST)↓ | Ang Err↓ | Accel Err↓ | SUCD↓ |
| WHAM | CVPR2024 | Camera | training | 82.83/97.55 | - | 12.24 | 4.51 | - | 69.42/83.35 | - | 10.13 | **9.20** | - |
| GVHMR | SIGGRAPH Asia 2024 | Camera | training | 72.47/84.23 | - | 13.80 | 4.59 | - | 79.39/91.23 | - | 13.49 | 19.00 | - |
| LiveHPS | CVPR2024 | LiDAR | training | 59.30/73.12 | 100.81/109.09 | 13.10 | 6.18 | 4.97 | 57.81/71.27 | 97.11/103.10 | 10.44 | 12.58 | 6.75 |
| LiveHPS++ | ECCV2024 | LiDAR | training | 54.15/57.74 | 88.91/97.53 | 11.85 | 4.37 | 3.47 | 55.67/68.73 | 89.06/95.85 | 10.26 | 11.51 | **5.60** |
| FreeCap | AAAI2025 | LiDAR+Camera | training | 53.31/65.50 | 95.97/102.91 | 11.14 | 5.97 | 4.82 | 55.45/68.52 | 96.47/102.67 | 9.14 | 9.60 | 6.66 |
| **Ours**[†] | - | LiDAR | training | 59.71/73.12 | 85.14/94.11 | 13.72 | 4.99 | 3.91 | 54.82/67.95 | 81.23/87.42 | 9.93 | 10.05 | 5.63 |
| **Ours** | - | LiDAR+Camera | training | **47.46/57.84** | **75.16/81.65** | **9.94** | **3.01** | 3.83 | **42.59/52.09** | **73.50/77.53** | **5.71** | 10.18 | 5.74 |
| WHAM | CVPR2024 | Camera | novel | 108.16/124.52 | - | 19.89 | 4.49 | - | 85.50/101.44 | - | 10.42 | **9.25** | - |
| GVHMR | SIGGRAPH Asia 2024 | Camera | novel | 114.00/126.41 | - | 18.22 | 4.32 | - | 83.02/95.83 | - | 14.68 | 20.14 | - |
| FreeCap | AAAI2025 | LiDAR+Camera | novel | 57.37/69.61 | 99.20/106.15 | 11.74 | 6.08 | 4.96 | 56.24/69.25 | 96.45/102.46 | 9.17 | 9.59 | 6.58 |
| **Ours** | - | LiDAR+Camera | novel | **52.25/63.05** | **79.05/85.82** | **10.80** | **3.16** | 3.93 | **42.64/52.28** | **73.41/77.51** | **5.76** | 10.15 | 5.74 |

Table 1: Evaluation on large-scale FreeMotion and Human-M3 datasets. The "View" specification differentiates between performance when using training versus novel camera viewpoints. Entries marked with "-" correspond to metrics that are inapplicable to camera-based methods due to their inherent scale ambiguity. **Ours**[†] denotes our method using only a single LiDAR sensor. Note that **Ours**[†] achieves comparable performance to LiDAR-only methods, LiveHPS and LiveHPS++, while the full multi-modal version yields substantial gains, underscores the effectiveness of our calibration-free architecture in effectively aligning and fusing diverse data from multiple multi-modal sensors

motion globally consistent, which helps a lot with translation-stable metrics like J/V(PST) and gives us a clear edge on fast outdoor sequences. For example, on HumanM3, we optimize PST errors from 89.06/95.85 (LiveHPS++) down to 81.23/87.42. We might lag slightly on a few indoor metrics, but we handle fast and varied motion way better thanks to that global consistency.

In novel camera viewpoint scenarios, camera-based methods such as WHAM and GVHMR exhibit significant performance degradation relative to their training views, while the multi-modal method FreeCap suffers from errors in its predicted calibration matrices. By eliminating the calibration step, our approach achieves substantially improved performance, **validating the effectiveness of our calibration-free design.**

Figure 4 visualizes our method's advantages in challenging scenarios: In cases of sparse point clouds, the integration of image priors helps maintain plausible body shape where LiDAR-only methods fail. Conversely, in outdoor settings with poor image quality, the LiDAR point clouds provide reliable geometric information. These visual results complement the quantitative analysis, illustrating the practical benefits of our calibration-free, multi-modal fusion approach. Please refer to Appendix B for more visual results on trajectory estimation analysis and complex motion capture.

**Cross-domain Evaluation on Noisy and Fast-Motion Datasets.** Results in Table 2 further reinforce the generalizability of our method, which is training on data from AMASS (Mahmood et al., 2019) and FreeMotion (Ren et al., 2024b) and test on challenging outdoor long-range noisy dataset LiDARHuman26M (Li et al., 2022) and the fast-

| Methods | Dataset LiDARHuman26M (Li et al., 2022) | | | | | Dataset RELI11D (Yan et al., 2024) | | | | |
|---|---|---|---|---|---|---|---|---|---|---|
| | J/V Err(PS)↓ | J/V Err(PST)↓ | Ang Err↓ | Accel Err↓ | SUCD↓ | J/V Err(PS)↓ | J/V Err(PST)↓ | Ang Err↓ | Accel Err↓ | SUCD↓ |
| WHAM | 151.18/191.73 | - | 32.92 | 3.16 | - | 106.13/126.71 | - | 22.10 | 2.70 | - |
| GVHMR | 187.01/226.19 | - | 28.58 | **3.04** | - | 111.73/131.73 | - | 21.37 | 2.42 | - |
| LiveHPS | 160.70/199.68 | 224.17/247.18 | 29.15 | 9.94 | 8.45 | 84.29/104.24 | 100.90/113.20 | 24.03 | 3.47 | 5.48 |
| LiveHPS++ | 141.03/179.68 | **196.30**/221.71 | 26.23 | 4.67 | **5.20** | 78.45/95.41 | 88.10/102.43 | 22.28 | 1.68 | 2.09 |
| FreeCap | 160.87/202.42 | 224.35/249.65 | 28.53 | 10.16 | 9.65 | 80.49/98.47 | 98.88/109.36 | 23.60 | 3.48 | 5.53 |
| **Ours** | **132.38/166.22** | 199.00/**218.37** | **24.14** | 3.70 | 7.54 | **61.04/73.19** | **81.03/86.59** | **18.89** | **1.41** | **1.51** |

Table 2: Cross-Domain Evaluation on Noisy and Fast-motion Benchmarks: the outdoor noisy Li-DARHuman26M dataset and the indoor fast-motion RELI11D datasets. Our approach is trained on datasets AMASS and FreeMotion.

motion Dataset RELI11D (Yan et al., 2024). **The results highlight the robustness of our Noise-resistant Trajectory Tracker (Section 3.2) against environmental variations and noise.**

## 4.2 ABLATION STUDIES

To quantitatively dissect the contribution of each component in our methodology, we conduct thorough ablation studies. We first analyze the core components of the Unified Across-Sensor Motion Estimator. As evidenced in Table 4, the choice of feature space is crucial. Replacing our human-centric space alignment with a LiDAR-centric approach leads to a marked increase in error, underlining the importance of this design for calibration-free operation. Similarly, our proposed fault-tolerant fusion strategy proves more effective than simple concatenation or addition of features, validating its role in dynamically integrating information from an arbitrary number of sensors. The analysis of the Noise-resistant Trajectory Tracker reveals that its iterative refinement mechanism is essential for accurate global localization. The trajectory error progressively decreases as the number of iterations increases, with 3 iterations providing an optimal balance between accuracy and computational cost. This iterative process is key to mitigating the effects of sensor noise and occlusion, as motivated in Section 3.2. More ablations on single-modal performance analysis and qualitative results are provided in Appendix F.

## 4.3 REAL-TIME MOTION CAPTURE IN REAL-WORLD LARGE-SCALE SCENES

In large-scale real-world motion capture scenarios, a single sensor is often inadequate, making multi-sensor setups essential. However, existing approach's reliability on calibration among fixed-placed sensors greatly limited their practicability. Can-Cap's ability to robustly handle varying sensor numbers and flexible placements of sensors in a calibration-free manner enable more practical real-world applications. Further technical and visual details on real world deployment are provided in Appendix A and the supplemental video.

## 4.4 SENSOR SCALABILITY

Can-Cap demonstrates strong **sensor scalability**, meaning it can adapt to varying numbers of sensors without retraining or recalibration. We evaluate this in two complementary settings:

**Scaling down:** Our single model, trained with multiple sensors, maintains over 92% of the performance even when evaluated with fewer LiDARs or cameras than seen during training (Table 3). This property allows a motion capture system to remain flexible when only a subset of sensors is available, which is common in large-scale or resource-constrained deployments.

**Scaling up:** Conversely, we train the model on a minimal sensor setup and test it with additional sensors at inference time (Table 5). This scenario is practically motivated: while collecting large-scale data with dense sensor arrays is often costly, extra sensors may be available at test time to improve performance. Can-Cap can seamlessly leverage these additional sensors, resulting in consistent error reduction and validating the scalability of the framework.

Together, these two aspects demonstrate that Can-Cap is both robust and adaptable to diverse sensor configurations, supporting calibration-free deployment in real-world multi-sensor motion capture scenarios.

| LiDAR | | | Camera | | | | Dataset FreeMotion | | | | |
|---|---|---|---|---|---|---|---|---|---|---|---|
| 1 | 2 | 3 | 0 | 1 | 2 | 3 | J/V Err(PS)↓ | J/V Err(PST)↓ | Ang Err↓ | Accel Err↓ | SUCD↓ |
| ✓ | | | ✓ | | | | 59.71/73.12 | 85.14/94.11 | 13.72 | 4.99 | 3.91 |
| | | | | | | | 67.49/83.73 | 91.86/103.37 | 14.11 | 5.06 | 4.39 |
| ✓ | | | | ✓ | | | 49.55/60.01 | 77.05/83.43 | 10.39 | 4.75 | 3.94 |
| | | | | | | | 50.28/60.88 | 77.68/84.21 | 10.38 | 4.79 | 3.96 |
| ✓ | | | | | ✓ | | 47.39/57.50 | 75.26/81.27 | 9.88 | 4.68 | 3.93 |
| | | | | | | | 45.98/55.50 | 74.17/79.72 | 9.45 | 4.64 | 3.83 |
| ✓ | | | | | | ✓ | 44.30/53.75 | 72.89/78.24 | 9.10 | 4.58 | 3.86 |
| | | | | | | | 44.40/53.54 | 72.93/78.16 | 9.09 | 4.58 | 3.79 |
| ✓ | ✓ | | ✓ | | | | 50.80/62.01 | 77.90/84.91 | 10.99 | 4.80 | 3.92 |
| | | | | | | | 49.79/60.28 | 72.87/79.43 | 10.25 | 4.17 | 4.30 |
| | ✓ | ✓ | | | | | 49.54/60.47 | 76.88/83.57 | 10.61 | 4.77 | 3.87 |
| | | | | | | | 49.27/59.70 | 77.03/83.45 | 10.19 | 4.77 | 3.98 |
| | | ✓ | | | | ✓ | 43.71/52.70 | 72.60/77.74 | 8.89 | 4.57 | 3.83 |

Table 3: Sensor Robustness(**scaling down**). Our single model (Black), trained with multiple sensors (3 LiDARs + 3 Cameras), can effectively adapt to various degraded sensor configurations at test time (e.g., using fewer LiDARs or cameras) while retaining over 92% of the performance of models trained specifically for those configurations (Gray).

| | | Dataset FreeMotion | | | | |
|---|---|---|---|---|---|---|
| | | J/V Err(PS)↓ | J/V Err(PST)↓ | Ang Err↓ | Accel Err↓ | SUCD↓ |
| Feature Space (UAME) | LiDAR-Centric | 68.73/81.42 | 92.38/100.81 | 12.75 | 3.39 | 5.11 |
| | Human-Centric | 52.25/63.05 | 78.87/85.66 | 10.80 | 3.12 | 3.91 |
| Feature Fusion (UAME) | cat | 60.63/72.41 | 85.49/93.09 | 11.73 | 3.27 | 4.45 |
| | add | 61.37/73.13 | 85.97/93.55 | 11.70 | 3.28 | 4.53 |
| | ours | 52.25/63.05 | 78.87/85.66 | 10.80 | 3.12 | 3.91 |
| Trajectory Estimation (NTT) | repeat × 1 | 52.25/63.05 | 87.88/94.01 | 10.80 | 5.05 | 4.26 |
| | repeat × 2 | 52.25/63.05 | 82.42/88.78 | 10.80 | 4.18 | 3.54 |
| | repeat × 3 | 52.25/63.05 | 78.87/85.66 | 10.80 | 3.12 | 3.91 |
| | repeat × 4 | 52.25/63.05 | 74.88/82.20 | 10.80 | 3.17 | 3.81 |
| | repeat × 5 | 52.25/63.05 | 74.63/82.00 | 10.80 | 3.17 | 3.75 |

Table 4: Ablation studies on our design choices. We analyze the impact of feature space, fusion strategy, trajectory estimation iterations. Note that the local pose metrics (J/V Err(PS) and Ang Err) remain unchanged across different trajectory estimation settings in Noise-resistant Trajectory Tracker (NTT), as this ablation module affect global translation only while maintaining the estmiated local pose fixed.

| | | Dataset FreeMotion | | | | |
|---|---|---|---|---|---|---|
| Train Sensors | Infer Sensors | J/V(PS) | J/V(PST) | Ang Err | Accel Err | SUCD |
| 2C+2L | 2C+2L | 47.24/57.45 | 70.58/76.98 | 6.75 | 4.02 | 3.74 |
| 2C+2L | 2C+3L | 47.40/57.65 | 67.51/73.93 | 6.73 | 3.87 | 3.79 |
| 2C+2L | 3C+2L | 45.75/55.67 | 69.38/75.50 | 6.56 | 3.96 | 3.68 |
| 2C+2L | 3C+3L | 45.92/55.87 | 66.27/72.41 | 6.54 | 3.81 | 3.72 |
| 1C+1L | 1C+1L | 49.55/60.01 | 77.05/83.43 | 10.39 | 4.75 | 3.94 |
| 1C+1L | 2C+1L | 47.34/58.12 | 72.20/81.90 | 9.51 | 4.78 | 3.50 |
| 1C+1L | 3C+1L | 46.33/59.92 | 74.37/80.90 | 9.40 | 4.73 | 3.47 |
| 1C+1L | 1C+2L | 49.02/62.97 | 72.24/79.15 | 9.65 | 4.22 | 3.77 |
| 1C+1L | 1C+3L | 48.94/62.81 | 69.15/76.10 | 9.63 | 4.08 | 3.86 |
| 1C+1L | 2C+3L | 46.86/60.38 | 67.32/73.94 | 9.41 | 3.98 | 3.75 |
| 1C+1L | 3C+3L | 46.12/59.50 | 66.70/73.19 | 9.32 | 3.94 | 3.72 |

Table 5: Sensor scaling-up experiments on the FreeMotion dataset. Models are trained on a minimal sensor setup (first column: **Train Sensors**) and evaluated with additional sensors at test time (second column: **Infer Sensors**) without retraining. Here, **C** denotes the number of cameras and **L** denotes the number of LiDARs. The consistent reduction in errors when adding sensors demonstrates that the learned representation generalizes gracefully to additional cameras and/or LiDARs, validating the scalability of Can-Cap.

## 5 CONCLUSIONS

To conclude, we present *Can-Cap*, a calibration-free motion capture framework that operates robustly with arbitrary combinations of LiDARs and cameras. It eliminates calibration through a Unified Across-Sensor Motion Estimator that aligns data in a human-centric space, and ensures noise robustness via a Noise-Resistant Trajectory Tracker with iterative refinement. Extensive experiments across multiple datasets and in the real world demonstrate state-of-the-art performance, validating the practicality and flexibility of our approach for real-world motion capture.

ETHICS STATEMENT

Our work presents no direct ethical concerns. The primary application of our method is for novel view synthesis from captured data.

REPRODUCIBILITY STATEMENT

To ensure reproducibility: (1) While not included with this submission, our full project (including the Unreal Engine renderer, training, and evaluation scripts) will be released on GitHub upon publication. (2) All experimental details are provided in Appendix D.1. (3) Our data and pre-processing follow the official LiveHPS++ and WHAM implementation. (4) Network details and computational efficiency of our method are provided in Appendix D.2.

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

# APPENDIX

## A DEPLOYMENT AND REAL-TIME IMPLEMENTATION

### A.1 REAL-TIME MOTION CAPTURE FRAMEWORK

Our method enables real-time motion capture by combining accurate local representations (pose $\hat{\beta}$ and shape $\hat{\theta}$ from Section 3.1) with global translation representations $\hat{\Gamma}$ (Section 3.2). While reconstructed SMPL parameters effectively represent human motion, online streaming scenarios demand enhanced temporal consistency and trajectory smoothness for applications such as biomechanics analysis, sports feedback, and real-time avatar control.

To address these requirements, we further introduce the *Online Trajectory Polisher (OTP)*, which enhances motion coherence and stability in frame-by-frame predictions, ensuring reliable performance in dynamic, unconstrained environments.

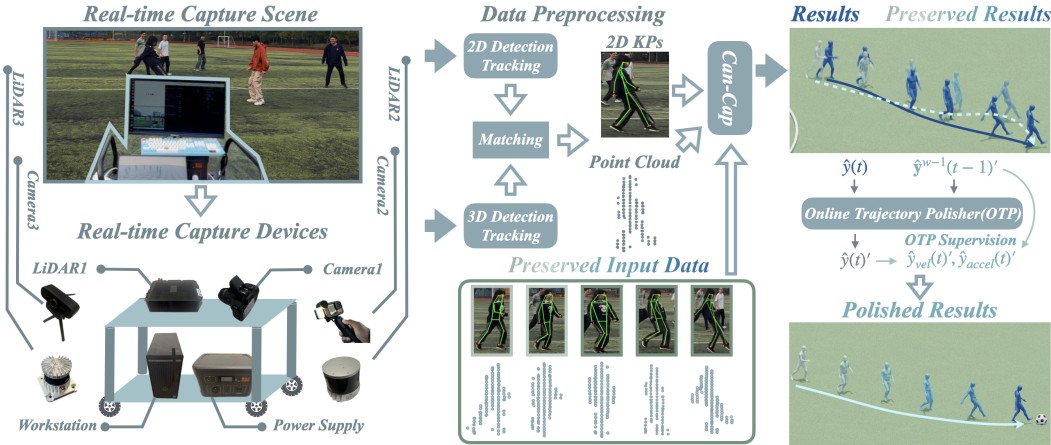

Figure 5: Architecture of Can-Cap-based real-time motion capture system. Left: Capture scenes and sensor configurations. Middle: Data preprocessing pipeline including detection, tracking, and matching. Right: Online Trajectory Polisher for post-optimization.

### A.2 ONLINE TRAJECTORY POLISHING STRATEGY

Previous methods (Ren et al., 2024a;b) employ a cache window preserving $w$ frames of input data to generate $w$ predictions $\hat{\mathbf{y}}^w(t) = [\hat{y}(t - w + 1), \hat{y}(t - w + 2), \cdots, \hat{y}(t)]$, selecting the final prediction $\hat{y}(t)' = \hat{y}(t)$ at each time step. However, this approach neglects information from previous predictions, leading to motion incoherence in real-time scenarios.

Our OTP strategy is motivated by the observation that global trajectories evolve smoothly while local poses may exhibit abrupt accelerations. We therefore apply selective smoothing to global trajectories while preserving local motion details. The polisher refines predictions by leveraging both historical predictions $\hat{\mathbf{y}}^{w-1}(t - 1)'$ and current window predictions $\hat{\mathbf{y}}^w(t)$:

$$\hat{y}(t)' = E_s(\hat{\mathbf{y}}^{w-1}(t-1)', \hat{y}(t)) \tag{8}$$

where $E_s$ represents a network with multiple temporal convolutions (kernel size $w$). We supervise training using comprehensive loss functions:

$$\mathcal{L}_{mse}(y(t)) = \|y(t) - \hat{y}(t)'\|_2^2 + \|y_{vel}(t) - \hat{y}_{vel}(t)'\|_2^2 + \|y_{accel}(t) - \hat{y}_{accel}(t)'\|_2^2 \tag{9}$$

where $y_{vel}(t)$ and $y_{accel}(t)$ denote velocity and acceleration components, respectively.

## A.3 DEPLOYMENT OPTIMIZATION

As depicted in Figure. 5, our method achieves real-time performance at 25 FPS on single LiDAR-camera configurations through TensorRT-accelerated processing pipelines, while supporting flexible sensor expansions. This processing speed exceeds the 10 FPS hardware limitation of our LiDAR sensors, ensuring full utilization of incoming data streams.

We enhance the matching process through several optimizations:

- **Matching Memory Bank**: Preserves identities of successfully matched individuals across sensors, reducing unnecessary computations
- **Error Accumulation**: Accumulates re-projection errors for both matched and unmatched pairs, providing prior knowledge for algorithm refinement
- **Hardware Acceleration**: Deployment of YoloX, ViTPose, and OSNet models with TensorRT-FP16 optimization
- **Custom CUDA Operators**: Implementation of furthest point sampling and point cloud clustering for improved efficiency
- **ReID Model**: We adopt OSNet as the ReID model for the RGB modality to mitigate frequent ID switches in crowded scenarios.

Our framework has been successfully deployed in real-world scenarios including tennis motion analysis for biomechanical evaluation and human-robot collaboration systems, demonstrating versatility for applications in immersive VR/AR training, clinical rehabilitation monitoring, and large-scale sports analytics.

## B QUALITATIVE EVALUATIONS

### B.1 TRAJECTORY ESTIMATION ANALYSIS

This section presents a case study on trajectory estimation, particularly focusing on fast motion scenarios. Figure. 6 reveals significant discrepancies in LiveHPS++'s translational accuracy during rapid movements, whereas our method maintains close alignment with point cloud data, demonstrating superior robustness and precision.

Figure. 7 compares long-term trajectory estimation with camera-based methods WHAM and GVHMR. Our method clearly outperforms alternatives, especially in capturing extended trajectories of running subjects. For visualization consistency, we align the first frame of all methods and scale meshes by a factor of 5.

### B.2 COMPLEX MOTION CAPTURE

This section demonstrates our method's capability in handling intricate human movements that challenge traditional motion capture. As shown in Figure. 8, our approach successfully captures complex motions including somersaults, plank poses, and dance movements. These visualizations highlight the advanced integration of LiDAR and camera data, showcasing our method's ability to accurately track and reconstruct dynamic human behaviors with high fidelity.

## C DATASETS

We provide comprehensive details of the datasets employed in our study. The selected datasets exhibit diversity in sensing modalities, environmental conditions, and motion complexity, ensuring thorough validation of our method.

- **Human-M3**: This outdoor multi-view multi-modal dataset captures unconstrained multi-person interactions across basketball courts and plazas. It provides synchronized LiDAR point clouds and 4-view RGB streams (12,200 frames) with optimization-based SMPL annotations. The dataset specifically addresses challenges of distant subjects (exceeding

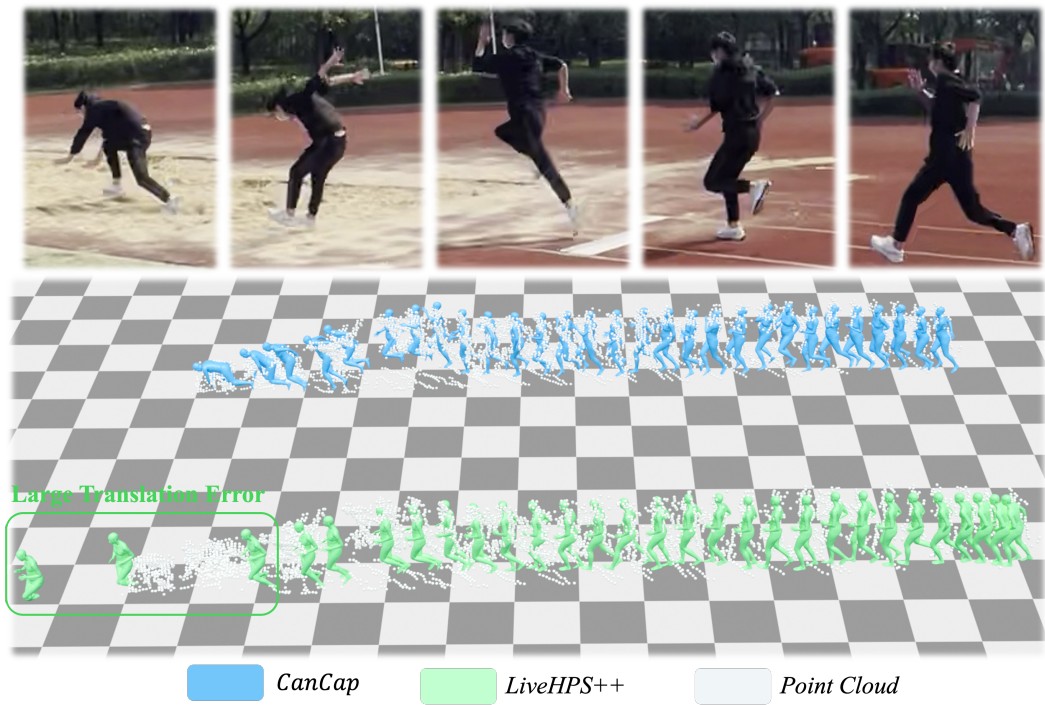

Figure 6: Qualitative comparison of fast motion capture, highlighting LiveHPS++'s translation errors in jumping scenarios. Our method maintains accurate alignment even under rapid motion conditions.

10m LiDAR distance) and heavy inter-person occlusion (38% samples with more than 50% occlusion). As shown in Figure. 10, Human-M3 includes cases with extreme lighting conditions, demonstrating our method's capability for day-and-night operation. The setup of Human-M3 is visualized in Figure. 9.

- **FreeMotion**: A large-scale indoor dataset captured in diverse real-life scenarios with natural occlusions and sensor noise. It encompasses multi-modal data including LiDAR point clouds, RGB images, and IMU measurements, totaling 578,775 frames with 1 to 7 performers per scene. The dataset provides high-precision full SMPL parameter annotations through a dense-camera capture system and LiDARs, offering rich resources for exploring various human motion capture configurations. The setup of FreeMotion is visualized in Figure. 9.

- **LiDARHuman26M**: A cross-dataset benchmark focusing on long-range outdoor poses (15-30m) captured by static LiDAR. While limited to single subjects, its precise IMU-based annotations and extreme sparsity (average 150 points/person) rigorously test robustness to incomplete geometric observations.

- **RELI11D**: A high-speed motion dataset covering five sports disciplines with four synchronized modalities: 128-beam LiDAR (20Hz), event camera (microsecond resolution), RGB (60Hz), and 17-IMU suits. Its 239K frames emphasize dynamic accuracy through contact-aware optimization losses and provide millimeter-accurate scene meshes for physics-based evaluation.

# D NETWORK AND IMPLEMENTATIONS

## D.1 IMPLEMENTATION DETAILS

Our framework is implemented with PyTorch 2.1.0 and CUDA 12.1. We employ a two-phase training strategy: initial pre-training on the SURREAL dataset (Varol et al., 2017) with generated 2D

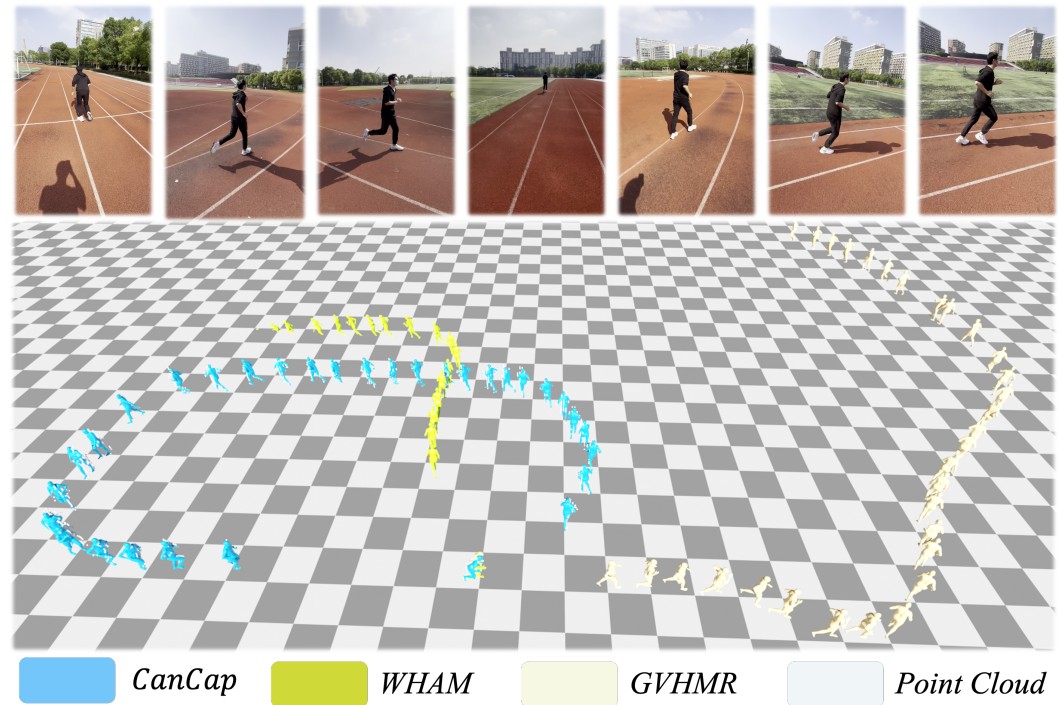

Figure 7: Qualitative comparison of long-term trajectory estimation. Our method accurately captures running trajectories where WHAM and GVHMR exhibit significant drift. Human meshes are scaled by $5\times$ for visualization clarity.

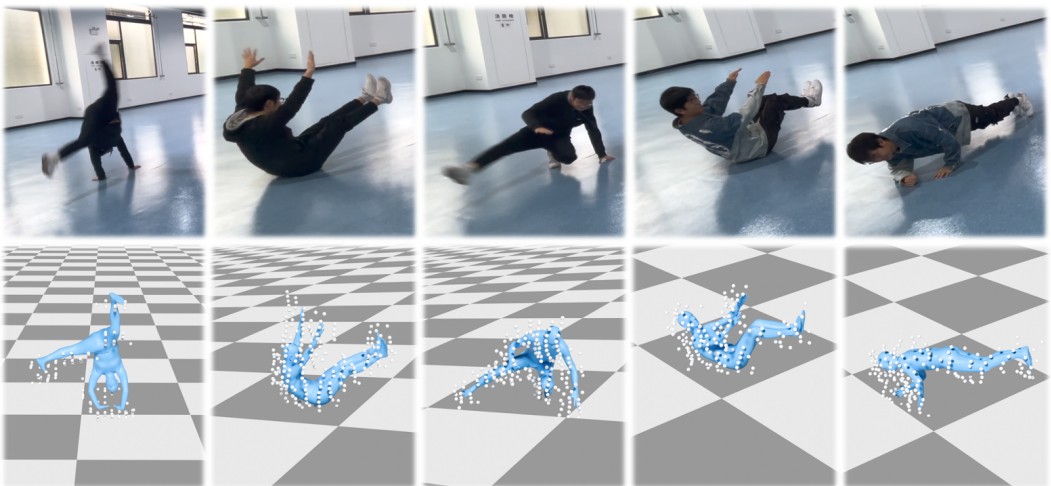

Figure 8: Qualitative results of complex motion capture demonstrating robust performance across diverse challenging scenarios including acrobatics, extreme poses, and dynamic movements.

keypoints from multiple camera views for 500 epochs, followed by fine-tuning on Human-M3 (Fan et al., 2023a) and FreeMotion-indoor (Ren et al., 2024a) datasets for 20 epochs with camera view augmentation. Training configurations include: batch size of 32, initial learning rate of $5 \times 10^{-4}$, AdamW optimizer with weight decay $10^{-4}$, and cosine learning rate scheduler. Hyperparameters are set as $\alpha = 0.01$, $\lambda = 0.1$, $\delta = 0.3$ and N=3.

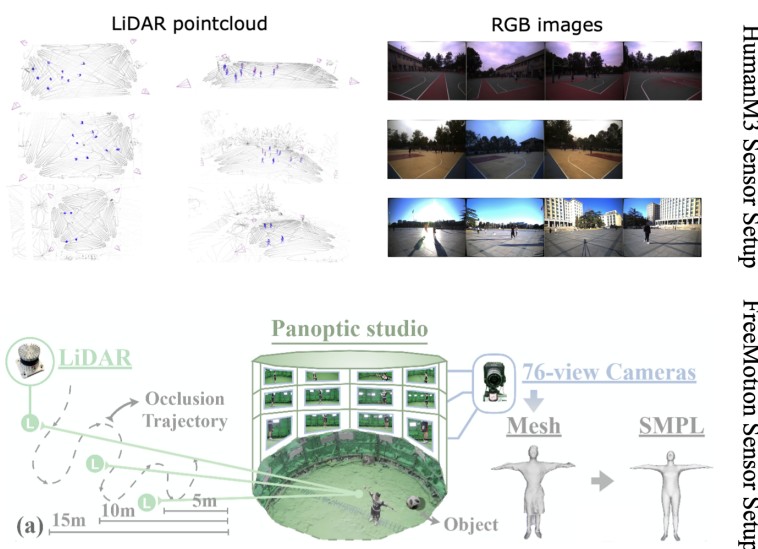

Figure 9: Sensor Setup of Human-M3 dataset and FreeMotion dataset.

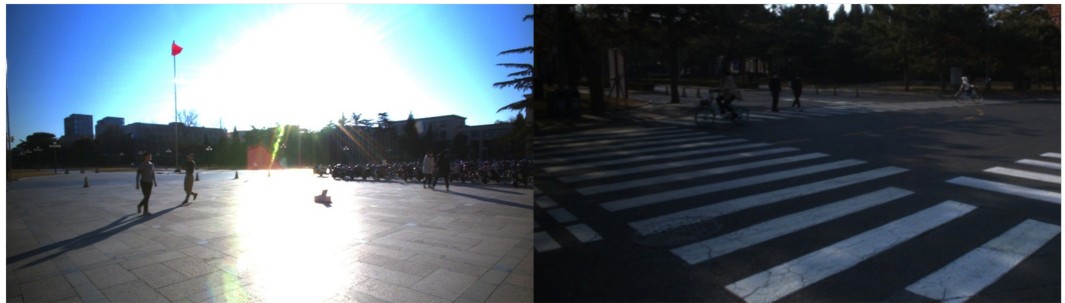

Figure 10: Representative cases from Human-M3 dataset demonstrating extreme lighting conditions. Our method maintains robust performance under both daytime and nighttime scenarios.

## D.2 NETWORK ARCHITECTURE

Details on architecture of the Unified Across-sensor Motion Estimator is shown in Table 6.

| Component | Layers | Input/Output | Parameters |
|---|---|---|---|
| **Human-Centric Space Alignment** | | | |
| LiDAR Processing | PointNet-GRU | $(B,N,T,256,3) \rightarrow (B,N,T,24,3)$ | 3 GRU layers, 512 units |
| Rotation Prediction | GRU | $(B,N,T,24,3) \rightarrow (B,N,T,6)$ | 3 layers, hidden=512 |
| Space Transformation | Rotation matrix | $(B,N,T,24,3) \rightarrow (B,N,T,24,3)$ | $R(\hat{\tau}_i)^{-1}$ |
| **Feature Encoder** | | | |
| 3D Encoder | GRU | $(B,N,T,24,3) \rightarrow (B,N,T,72)$ | 3 layers, 512 units |
| 2D Encoder | GRU | $(B,M,T,17\times2+3) \rightarrow (B,M,T,563)$ | 3 layers, hidden=512 |
| **Fault-Tolerant Sensor Fusion** | | | |
| Cross-Modal Fusion | Transformer | $(72+563)D \rightarrow 512D$ | 4 heads, 2 layers |
| SMPL Decoder | GRU | $512D \rightarrow (24\times6+10)D$ | 3 layers, hidden=512 |

Table 6: Detailed architecture specifications of Unified Across-sensor Motion Estimator
B: batch size, N: LiDAR count, M: camera count, T: sequence length.

| Method | WHAM | GVHMR | LiveHPS | LiveHPS++ | FreeCap | Ours |
|---|---|---|---|---|---|---|
| FPS | 8/- | 5/- | 51/191 | 26/162 | 8/15 | 12/25 |
| Parameters (M) | 42 | 40 | 46 | 104 | 53 | 116 |

Table 7: Computational efficiency comparison with state-of-the-art methods. Each entry reports PyTorch FPS / TensorRT FPS. WHAM and GVHMR do not provide deployable TensorRT implementations due to their complex multi-stage architectures(for example offline SLAM model), thus only PyTorch FPS is reported.

| LiDAR | | | | Camera | | | | FreeMotion | | | | |
|---|---|---|---|---|---|---|---|---|---|---|---|---|
| 0 | 1 | 2 | 3 | 0 | 1 | 2 | 3 | J/V Err(PS)↓ | J/V Err(PST)↓ | Ang Err↓ | Accel Err↓ | SUCD↓ |
| | ✓ | | | | ✓ | | | 59.71/73.12 | 85.14/94.11 | 13.72 | 4.99 | 3.91 |
| | ✓ | | | | ✓ | | | 67.49/83.73 | 91.86/103.37 | 14.11 | 5.06 | 4.39 |
| | | ✓ | | | ✓ | | | 56.61/68.89 | 82.62/90.71 | 12.62 | 4.92 | 3.96 |
| | | ✓ | | | ✓ | | | 65.33/81.34 | 90.12/101.44 | 13.76 | 5.00 | 4.36 |
| | | ✓ | ✓ | | | | | 54.54/66.59 | 80.84/88.58 | 11.96 | 4.90 | 3.92 |
| | | ✓ | ✓ | | | | | 64.50/80.68 | 89.42/100.87 | 13.65 | 4.97 | 4.39 |
| ✓ | | | | | ✓ | | | 71.75/85.94 | - | 13.45 | - | - |
| ✓ | | | | | ✓ | | | 75.34/89.16 | - | 13.37 | - | - |
| ✓ | | | | | | ✓ | | 58.02/69.35 | - | 10.82 | - | - |
| ✓ | | | | | | ✓ | | 64.18/75.11 | - | 11.09 | - | - |
| ✓ | | | | | | | ✓ | 53.33/63.62 | - | 10.16 | - | - |
| ✓ | | | | | | | ✓ | 60.73/70.89 | - | 10.31 | - | - |

Table 8: Ablation study evaluating Can-Cap performance under various sensor configurations

# E    COMPUTATIONAL EFFICIENCY

Table 7 benchmarks our network against state-of-the-art methods in real-time performance and model complexity. Despite increased parameter count (116M) due to multi-sensor fusion capabilities, our architecture achieves 25 FPS—3× faster than WHAM/GVHMR. This efficiency stems from TensorRT acceleration and optimized GRU-based recurrence, enabling real-time operation on edge devices. The parameter increase primarily resides in the cross-modal transformer (12.7% of total parameters), justified by its critical role in calibration-free fusion. However, the current bottleneck of our method lies in detection/tracking and cross-sensor matching.

# F    ABLATION STUDIES

## F.1    SINGLE-MODAL PERFORMANCE ANALYSIS

To validate the robustness of our framework under different sensor configurations, we conduct comprehensive ablation studies using individual sensor modalities. Table 8 demonstrates that our model maintains reasonable performance even when relying solely on either LiDAR or camera inputs. Gray-colored results represent models trained individually under specific settings, while non-colored results stem from a unified model trained with 3 LiDARs and 3 cameras, assessed by masking sensor inputs to align with corresponding configurations.

## F.2    ABLATION ANALYSIS ON NOISE LEVELS

To systematically evaluate the effectiveness of NTT under varying noise conditions, we constructed three object-noise subsets of the FreeMotion dataset by adding synthetic clutter around human point clouds. The subsets are defined as follows:

- **Level 1 (★☆☆):** A subset of the original dataset containing approximately 1/3 of samples with minimal noise.

- **Level 2 (★★☆):** Objects with a maximum side length of 0.1-0.5 m, sparsely placed around the human.
- **Level 3 (★★★):** Larger objects with a maximum side length of 0.5-1 m, creating moderate surrounding clutter.

The synthetic object noise is generated as follows:

1. We sample an object from ShapeNet and scale it to the target size corresponding to the noise level.
2. The object is placed in a virtual scene sharing the same viewpoint as the human point cloud(real LiDAR's viewpoint).
3. We perform raycast simulation from the point cloud viewpoint to generate the object points visible from the sensor perspective.
4. The resulting object point cloud is then randomly positioned around the human to create the noisy sample.

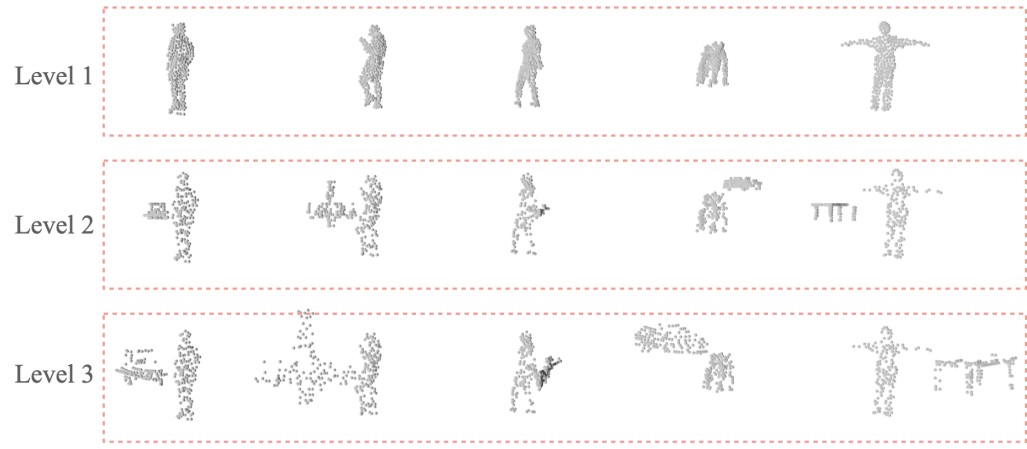

Figure 11: Examples of the three noise levels: top row shows Level 1, middle row Level 2, and bottom row Level 3.

Figure 11 shows representative examples of the three noise levels. We evaluate NTT by comparing its performance with and without NTT(N=3). The results are summarized in Table 9.

| Noise Level | Setting | J/V(PS) | J/V(PST) | Ang Err | Accel Err | SUCD |
|---|---|---|---|---|---|---|
| ★☆☆ | w/o NTT | 38.75/46.54 | 67.64/72.45 | 6.93 | 3.60 | 3.33 |
| | w/ NTT | 38.75/46.54 | 64.01/68.98 | 6.93 | 2.89 | 3.86 |
| ★★☆ | w/o NTT | 50.47/59.94 | 96.49/101.84 | 7.80 | 10.54 | 189.51 |
| | w/ NTT | 50.47/59.94 | 89.49/95.33 | 7.80 | 3.08 | 185.45 |
| ★★★ | w/o NTT | 84.17/100.98 | 367.51/374.16 | 8.67 | 94.62 | 553.86 |
| | w/ NTT | 84.17/100.98 | 158.53/165.00 | 8.67 | 6.71 | 377.83 |

Table 9: Ablation study of NTT under varying noise levels. Level 1 (★☆☆) uses low-noise original point clouds, while Levels 2 (★★☆) and 3 (★★★) include synthetic object noise generated from ShapeNet models as described in the text. Across all noise levels, NTT consistently reduces trajectory error J/V(PST) and temporal inconsistency (Accel Err), particularly under challenging noisy conditions.

As shown, NTT effectively mitigates the negative impact of noise. The largest improvements occur in Level 3, where surrounding clutter is significant, demonstrating the robustness of NTT under adverse conditions.

### F.3 QUALITATIVE ABLATION ANALYSIS

We provide qualitative ablation studies to demonstrate our method's robustness across different components. Figure. 13 illustrates that the Human-Centric Space outperforms LiDAR-Centric Space in maintaining robustness against camera perspective variations. Figure. 14 showcases the effectiveness of our feature fusion method in reconstructing human meshes under noisy sensor conditions. Finally, Figure. 15 demonstrates the progressive improvement of our Noise-resistant Trajectory Tracker across multiple iterations, showing enhanced alignment between point clouds and human meshes.

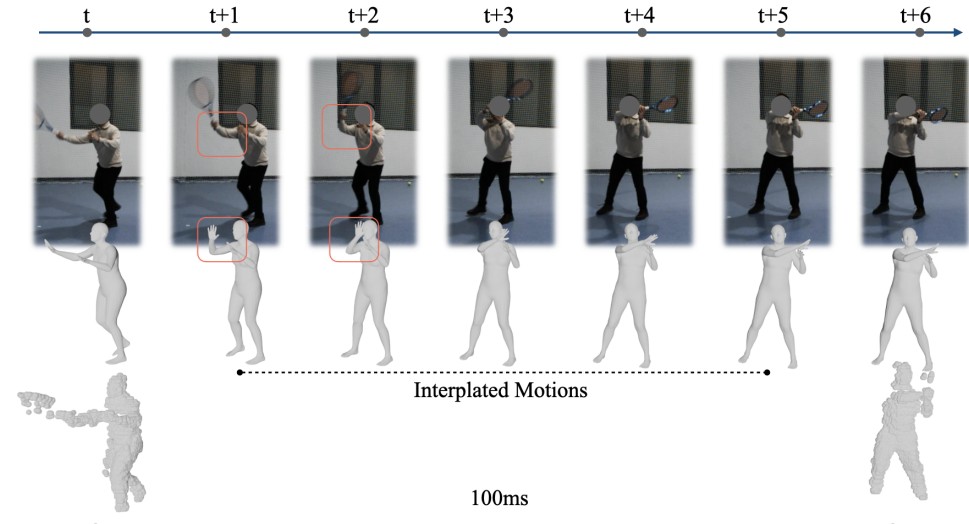

Figure 12: Temporal alignment protocol. Camera images (top) are dowm-sampled to 10 Hz to match LiDAR point clouds(bottom); meshes (middle) are reconstructed only on LiDAR frames and linearly interpolated elsewhere. Interpolation artifacts become evident under rapid, non-linear motion.

### F.4 MORE RESULTS ON RELI11D BENCHMARK

| Method | ACCEL↓ | MPJPE↓ | PA-MPJPE↓ | PVE↓ | PCK$_{0.3}$ ↑ |
|---|---|---|---|---|---|
| ImmFusion | 49.19 | 175.00 | 159.62 | 187.31 | 0.67 |
| FusionPose | 44.89 | 136.15 | 110.19 | 166.94 | 0.75 |
| LEIR | 27.07 | 55.36 | 45.72 | 122.32 | 0.90 |
| Ours | 20.40 | 50.47 | 43.61 | 64.92 | 0.95 |

Table 10: Quantitative comparison against state-of-the-art methods. Metrics are evaluated using the code of RELI11D benchmark.

Although our method operates in a *calibration-free* multi-modal fusion regime while the competing LiDAR+camera approaches rely on accurate extrinsic calibration. we nevertheless report results on the RELI11D benchmark to demonstrate the superiority of our framework. All baselines in Table 10 are multi-modal LiDAR-Camera method. We keep the native point-cloud representation, eliminating voxel artifacts, and perform robust fusion that fully exploits the complementary strengths of geometry and appearance.

## G LIMITATIONS AND FUTURE WORK

While our study demonstrates promising results, we acknowledge several limitations that present opportunities for future research:

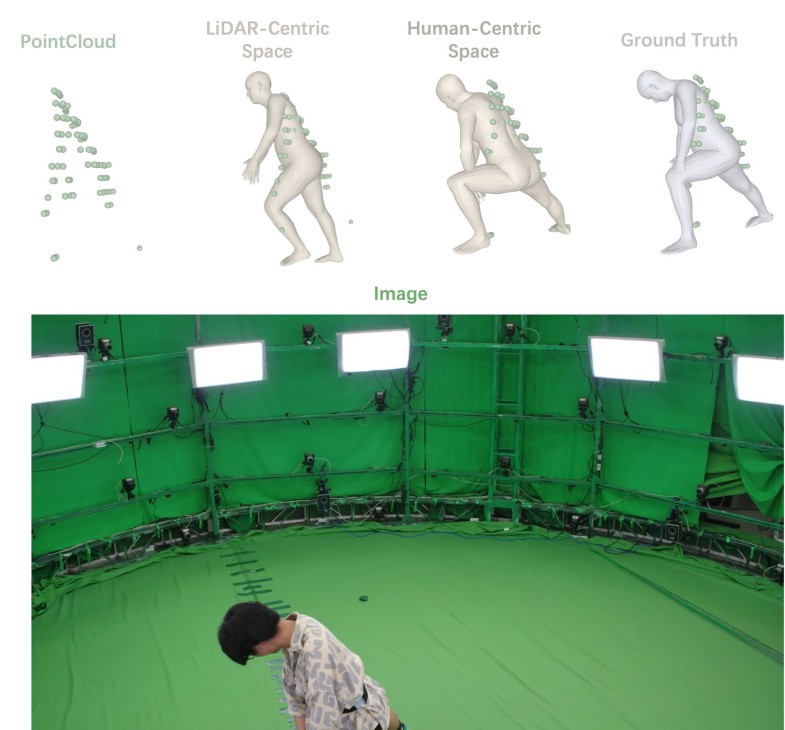

Figure 13: Qualitative comparison of feature space representations: Human-Centric Space (right) demonstrates superior robustness to perspective changes compared to LiDAR-Centric Space (left)

### G.1 SENSOR FREQUENCY MISMATCH

The LiDAR operates at 10 Hz while the cameras stream at 30 or higher FPS. To match them, we dowm-sample the camera frame rate to 10 Hz. To ensure the motion appears visually fluid, we upsample the motion results to higher FPS(60Hz as in our supplemented demo). However, extreme fast motions such as tennis may lose intermediate frame information, resulting in inaccurate motions; the resulting pose is accurate at the LiDAR frames but the interpolated motions are only a linear approximation of pose and translation. As Figure. 12 shows, because the tennis swing motion is highly non-linear, the interpolated poses are not accurate. Future work will retain the full high FPS camera stream and treat the 10 Hz LiDAR as a spatio-temporal prior, learning a latent motion model that can hallucinate the missing high-frequency dynamics instead of simply interpolating them.

### G.2 DEPENDENCY ON TRACKING ACCURACY

Can-Cap assumes that every input frame is equipped with a reliable human bounding box and a consistent identity label. In practice, however, both the 2D keypoint extractor and the cross-modal person-matching module are fed by off-the-shelf detectors and Re-ID networks whose failures are not recovered by the subsequent pose stage. This dependency becomes critical in the LiDAR branch: LiDAR data lack texture information, making Re-ID extremely challenging. Instead of treating perception as a pre-processing step, a promising direction is to jointly model human pose estimation and human tracking: the point cloud and image of the next frame is matched against the surrounding motion primitives from the previous frame and directly regresses the poses, instead of first tracking the point cloud and then performing association and pose estimation.

### G.3 GENERALIZATION TO EXTREME CONDITIONS

While robust to various lighting conditions, performance in severe weather conditions (heavy rain, fog) requires further validation. Future iterations could incorporate weather-resistant sensor configurations and corresponding training data augmentation.

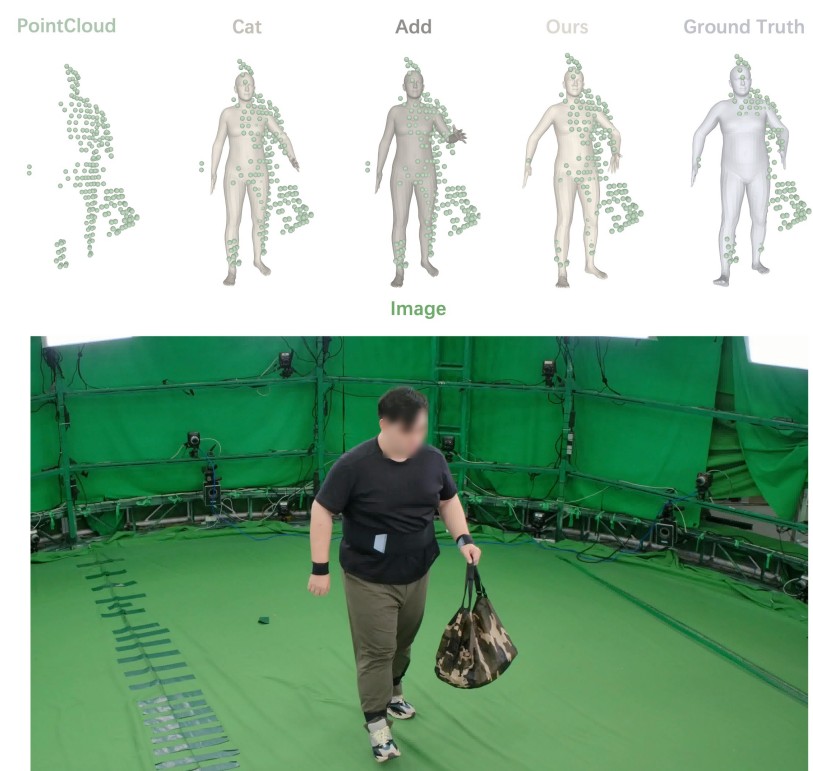

Figure 14: Ablation study of feature fusion strategies under noisy sensor conditions

## USAGE OF LARGE LANGUAGE MODELS

During the preparation of this work, the authors used LLM solely for the purpose of improving language fluency and checking grammatical errors in previously written drafts. After using this tool, the authors review and edit the content as needed and take full responsibility for the intellectual content of the publication.

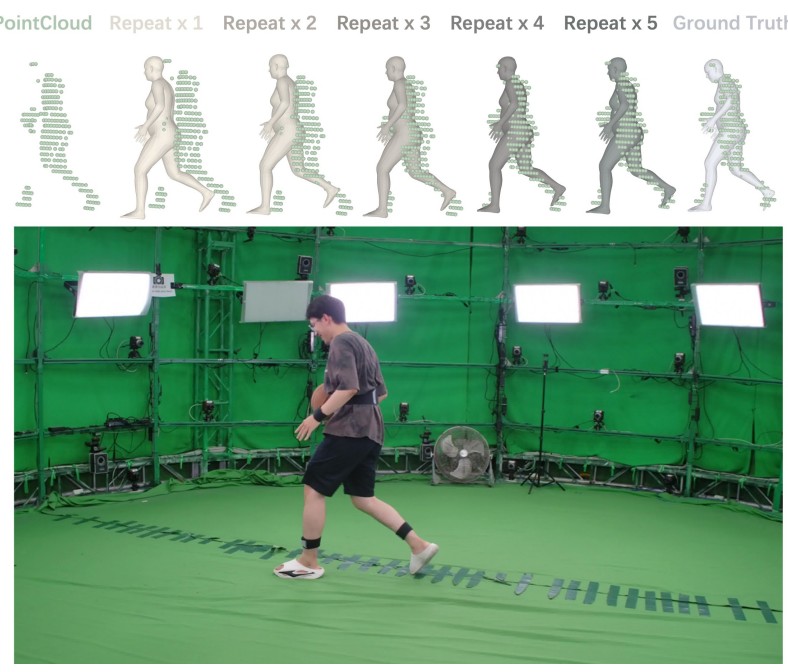

Figure 15: Progressive improvement of trajectory estimation through iterative refinement with our Noise-resistant Trajectory Tracker

