# OpenReview forum: "Can-Cap: Calibration-Free and Noise-Resilient Human Motion Capture via LiDAR-Camera Integration"
_ICLR.cc/2026/Conference — Submitted to ICLR 2026_

### Official Review · Reviewer_wiYn · 2025-10-29

**Soundness:** 3
**Presentation:** 2
**Contribution:** 2
**Rating:** 4
**Confidence:** 4

**Summary:**

This work proposes a calibration-free and noise-resilient 3D human motion capture framework. A key difference between this work and existing methods is that it can support a flexible number of sensors. Its performance on long-term trajectory estimation also significantly outperforms other methods.

**Strengths:**

1.The proposed method supports a flexible number of sensors and has been thoroughly evaluated through ablation studies under different sensor configurations.

2.It achieves improvements over existing methods such as WHAM, LiveHPS++, and FreeCap.

**Weaknesses:**

1.The overall contribution is rather limited. I believe this work simply combines the WHAM and LiveHPS++ methods, without significant innovation in the network architecture.
2.There is a lack of comparative experiments with other multimodal methods; the experiments include only FreeCap as a multimodal method.

**Questions:**

1.Could the authors specifically describe how multiple LiDARs and cameras were set up during data collection?
2.Could experiments be conducted on other datasets with multiple LiDARs and cameras?
3.Why does Human-Centric spatial alignment perform better than LiDAR-Centric spatial alignment?

---

> ### Author Response · Authors · 2025-11-20
>
> We thank the reviewer for the thorough assessment and insightful comments.
>
> ---
>
> ## W1) Core Innovations in Can-Cap vs. WHAM/LiveHPS++
> In Can-Cap, WHAM and LiveHPS++ serve as single-modality submodules; our **innovation lies in fusing them into a multi-modal, human-centric framework**. By projecting observations into a **human-centric frame**, Can-Cap is calibration-free, accommodating varying sensor placements. The **Noise-Tolerant Trajectory Tracker (NTT)** iteratively refines global translation, stabilizing motion under noisy LiDAR input, while **attention-based fusion** handles missing sensors automatically.
>
> **These designs address limitations WHAM/LiveHPS++ alone cannot solve: $\textcolor{red}{\text{viewpoint sensitivity}}$, $\textcolor{red}{\text{outdoor motion drift}}$, and $\textcolor{red}{\text{sensor failure}}$. By combining single-modality strengths in a multi-modal architecture, Can-Cap achieves robust, generalizable motion capture beyond simple module combination.**
>
> ## W2) Multi-Modal Comparisons and Q2) Multi-Modal Dataset Evaluation
> **W2).** Following your suggestions, **we compare with three more SOTA Multi-Modal Methods** on the RELI11D benchmark, which includes multi-modal(LiDAR+Camera) setups. But please note that **most, if not all, existing LiDAR+camera methods require calibration, while we don't. But we still achieve the best performance**.   Our approach fully exploits multi-modal data and significantly outperforms other methods. The metrics are aligned with original RELI11D benchmark using their code.
> | Method       | ACCEL↓ | MPJPE↓ | PA-MPJPE↓ | PVE↓  | PCK0.3↑ |
> |--------------|-----------|-----------|-----------|-------|---------|
> | ImmFusion(ICRA 2023)    | 49.19     | 175.00    | 159.62    | 187.31| 0.67    |
> | FusionPose(AAAI 2023)   | 44.89     | 136.15    | 110.19    | 166.94| 0.75    |
> | LEIR(CVPR 2024)         | 27.07     | 55.36     | 45.72     | 122.32| 0.90    |
> | **Ours**     | **20.40**     | **50.47**     | **43.61**     | **64.92** | **0.95**    |
>
> **Q2). More multi-modal Dataset.**
> To the best of our knowledge, **FreeMotion and Human-M3 are the only publicly available datasets** that provide multiple LiDARs and multiple cameras for human motion capture, **both of which we have tested on and achieved superior performance**.
> **Extended discussion of Generalization Capability**. Meanwhile, to further demonstrate the generalization of our approach, we additionally conduct **cross-dataset evaluation** on datasets **RELI11D** and **LiDARHuman26M** when training Can-Cap on dataset FreeMotion, which, while not multi-LiDAR setups, allow us to validate the model's performance under unseen environments and motion patterns, as shown in Table 2 of the main text.  These results validate that our method generalizes well to new domains.
>
> ## Q1) Sensor Setup of Dataset
> - **Training & Evaluation Datasets:**
>     - FreeMotion:  Dense capture with **76 synchronized RGB cameras** in a full arc and **3 LiDARs** at different heights. For ablations, we use a sparse **3-camera subset** at roughly **−120°, 0°, +120°**, simulating a balanced tri-view setup with minimal input. LiDARs scan at 10 Hz with overlapping coverage.
>     - Human-M3:  **3 LiDARs** and **3–4 RGB cameras** at outdoor corner positions forming wide-baseline views. Only **merged LiDAR point clouds** are provided, so the model sees a single unified point cloud per frame. Cameras operate at higher FPS and elevated positions to capture long-range, dynamic motions.
>
> We refer the reviewer to Appendix C for detailed sensor setups.
>
> ## Q3) Human-Centric vs. LiDAR-Centric Alignment
> We clarify that the difference between the two alignment strategies is not a minor or optional choice; it is a **fundamental design decision** that directly affects generalization across viewpoints (calibration-free).
>
> The core problem with a **LiDAR-centric representation** is its poor generalization. By anchoring features to a specific LiDAR, the model must implicitly learn extrinsic parameters that vary infinitely with viewpoint. This creates a learning space too vast for the training set to cover, resulting in the failure we observe on unseen views.
>
> A **human-centric representation** resolves this issue entirely. All observations are re-expressed in a body-anchored local frame, making the representation independent of where the sensors are physically located. Once projected into this shared human-centric space, the features become consistent across views, and the network only needs to learn **person-relative geometry and motion**, which reside in a compact and stable distribution that can be fully captured during training.
>
> Empirically, the human-centric representation's advantage is clear: it reduces MPJPE by 16 mm under novel-view evaluation compared to the LiDAR-centric approach (Table 4). This validates that by eliminating extrinsic variability, we achieve significantly lower complexity, and stronger generalization.

---

### Official Review · Reviewer_9aPy · 2025-10-30

**Soundness:** 3
**Presentation:** 2
**Contribution:** 2
**Rating:** 4
**Confidence:** 3

**Summary:**

The paper introduces Can-Cap, a framework for 3D human motion capture which integrates multi-modal data from LiDAR and cameras. The primary contributions are that it is “Calibration-Free” and “Noise-Resilient”, which the authors claim to be the key limitations of the existing multi-modal approaches. Can-Cap achieves this by two core components, the Unified Across-Sensor Motion Estimator (which reconstructs the local pose and shape) and the Noise-Resistant Trajectory Tracker (which estimates the global trajectory).

**Strengths:**

1. **Strong Empirical Results**: The method achieves strong results on the provided benchmarks (Human-M3 and FreeMotion), outperforming prior SOTA methods like FreeCap, WHAM, and LiveHPS++ in several key metrics, especially in novel viewpoint scenarios.

2. **Interesting Core Concept**: The "calibration-free" approach, enabled by the Unified Across-Sensor Motion Estimator (UAME) that maps inputs to a common human-centric space, is a promising direction that avoids the error propagation common in explicit calibration steps.

3. **Sensor Flexibility**: The paper demonstrates that a single model trained with multiple sensors (e.g., 3 LiDARs + 3 Cameras) can still function effectively when "degraded sensor configurations" (e.g., fewer sensors) are used at test time, which is a useful feature for real-world robustness.

**Weaknesses:**

The paper's primary weakness is that its central claims of "Fault-tolerance" and "Noise-resistance" are "over-sold" and not sufficiently supported by the explanations or experiments. The work is difficult to evaluate fully, as it lacks a clear analysis of its limitations and failure cases.
1. **Unclear "Fault-Tolerant" Claim**: The paper claims the UAME's second stage is a "Fault-Tolerant Sensor Fusion" module. However, the text does not explain what makes this module "fault-tolerant." It appears to be a standard attention-based sensor fusion mechanism. The motivation for using attention is weak, and the ablation study (Table 4) merely shows it outperforms simple concatenation or addition ("ours" vs. "cat" and "add"). This provides limited insight and does not substantiate the claim of fault tolerance.
2. **Insufficient Evidence for "Noise-Resistance"**: The paper's motivation for the Noise-Resistant Trajectory Tracker (NTT) is sound (i.e., standard normalization fails under noise). However, the description of the NTT itself lacks explanatory depth. It is not clear how the proposed iterative refinement (Algorithm on pg 6) is "noise-resistant" in principle, as it appears to be a standard iterative refinement technique. The experimental evidence—while showing good performance on the noisy LiDARHuman26M dataset (Table 2)—is not a direct proof of noise resistance. The evaluation would greatly benefit from a controlled experiment on a clean dataset where increasing levels of synthetic noise are introduced to measure the NTT's performance degradation. Without this, the "noise-resistant" claim is an assertion, not a proven property.
3. **Misleading "Sensor Scalability" Claim**: The "sensor scalability" claim is misleading. "Scalability" suggests the system can scale up to use more sensors during inference than were used at training time. The experiment in Table 3 demonstrates the opposite: the model shows graceful degradation when using fewer sensors than it was trained on. This is a useful feature (robustness to sensor drop-out), but it is not scalability.
4. **Unhelpful Figure**: Figure 2, while visually appealing, does little to help the reader understand the novelty of the UAME and NTT. It is a high-level data-flow diagram that includes unexplained elements, such as the "Dynamic Mask" and the "Searching Space," which are never mentioned or explained in the main text. This is a significant flaw in the paper's presentation.
5. **Lack of Failure Case Analysis**: The paper is lacking in its analysis of failure cases. The "Limitations" section in the appendix only briefly states that performance degrades in "severe weather conditions", which is a generic limitation. A more in-depth discussion of when the UAME's alignment fails or when the NTT's tracking breaks would be necessary to understand the method's true boundaries.

**Questions:**

1. What, specifically, makes the Stage 2 "Fault-Tolerant Sensor Fusion" module "fault-tolerant" beyond what a standard attention-based fusion module provides?
2. In Stage 2, the paper introduces a "learnable fusion token" ($F_{token}$) that "serves as a query to aggregate multi-modal features". Is this a single, shared token for the entire model? This token's role and design could be introduced more clearly.
3. Could you please explain what is architecturally "noise-resistant" in the NTT?
4. Can the authors please clarify if the model can "scale" in the other direction? For instance, can the same model trained on (N=3, M=3) sensors accept input from (N=4, M=4) sensors at inference time or is the number of input sensors capped after training?
5. Could you please explain what the "Dynamic Mask" and the "Searching Space" components are?

---

> ### Author Response · Authors · 2025-11-20
>
> We thank the reviewer for the thorough assessment and insightful comments. We have addressed the concerns below and updated the manuscript accordingly.
>
> ---
>
> ## W1,Q1,Q2) Motivation and Novelty of Fault-Tolerant Sensor Fusion
> We apologize that we did not state clearly in the original manuscript.
>
> **Motivation**: To support a varying number of uncalibrated sensors in a unified model, **Can-Cap** must tolerate missing information when certain sensors are absent, which we refer to as Fault Tolerance.
>
> The fault tolerance in UAME Stage 2 stems from 1) the **dynamic mask**(sensor dropout) and 2) the **learnable fusion token**  in cross-attention.
>
> 1) The **dynamic mask** is a **dropout** applied as a binary visibility mask on the entire sensor features during training. It ensures missing modalities contribute zero weight after softmax, dynamically reassigning attention to available modalities while preserving gradient consistency. Unlike static fusion (e.g., concatenation or addition) that cannot reweight missing channels, this attention-based design **enables tolerance to sensor drop-out** during inference. Experiments show maintained performance even with test sensor configurations differing from training. We renamed it as "sensor dropout" to make it easier for readers to understand.
>
> 2) The **learnable fusion token** is a **single global vector** that serves as the query to aggregate all concatenated modality features. It is **independent of sensor count**, compressing multi-modal features into a human-centric representation. Table 4 in the manuscript confirms that attention-based fusion outperforms concatenation or addition.
>
> ## W2, Q3) Noise-Resistance of Iterative Refinement
> The structural noise-resistance of NTT stems from **iterative small-step refinement** instead of single-step large regression. As shown in Figure 3, if external points shift the point cloud center by several meters, single-step regression would require predicting an unusually large translation beyond the clean sample distribution. In NTT, each iteration predicts a bounded step ($\Delta \Gamma \in [-\delta, \delta]^3$), shared across clean and noisy samples, and accumulates over \(N\) iterations to approach the true root translation. This transforms an unbounded prediction space into a bounded space, keeping network outputs within the training distribution.
> We supplement this with controlled experiments on FreeMotion under increasing noise levels. Please see Table 9 in the Appendix for complete results.
>
> ## W3, Q4) Sensor Up-Scalability and Down-Scalability
> We clarify that our method gracefully handles **fewer sensors** and **more sensors** at inference than during training.
> The table below shows models trained with 1–2 cameras and 1–2 LiDARs evaluated with up to 3 cameras and 3 LiDARs, validating up-scalability. Performance remains comparable, demonstrating flexibility to unseen sensor counts without retraining.
>
> **Tables for sensor Up-Scalability experiments:**
> |Train Sensors|Infer Sensors|J/V(PS)|J/V(PST)|Ang Err|Accel Err|SUCD|
> |---|---|---|---|---|---|---|
> |2C+2L|2C+2L|47.24/57.45|70.58/76.98|6.75|4.02|3.74|
> |2C+2L|2C+3L|47.40/57.65|67.51/73.93|6.73|3.87|3.79|
> |2C+2L|3C+2L|45.75/55.67|69.38/75.50|6.56|3.96|3.68|
> |2C+2L|3C+3L|45.92/55.87|66.27/72.41|6.54|3.81|3.72|
> |1C+1L|1C+1L|49.55/60.01|77.05/83.43|10.39|4.75|3.94|
> |1C+1L|2C+1L|47.34/58.12|72.20/81.90|9.51|4.78|3.50|
> |1C+1L|3C+1L|46.33/59.92|74.37/80.90|9.40|4.73|3.47|
> |1C+1L|1C+2L|49.02/62.97|72.24/79.15|9.65|4.22|3.77|
> |1C+1L|1C+3L|48.94/62.81|69.15/76.10|9.63|4.08|3.86|
> |1C+1L|2C+3L|46.86/60.38|67.32/73.94|9.41|3.98|3.75|
> |1C+1L|3C+3L|46.12/59.50|66.70/73.19|9.32|3.94|3.72|
>
> ## W4, Q5) Figure and Main Text Revision on "Dynamic Mask" and "Searching Space"
> We appreciate your advice and apologize for our mistake. We have revised the figure and main text accordingly. **All components corresponding to our key contributions are highlighted in orange.**
>
> The **Dynamic Mask** is technically a dropout mechanism that randomly drops entire sensor features during training to simulate sensor failure. It ensures masked modalities are excluded from attention aggregation, supporting fault tolerance. We renamed it as "sensor dropout" in the revised manuscript for clarity.
>
> The **Searching Space** in NTT defines the 3D spherical region allowed for iterative correction per step. Removing NTT would require direct regression over an unbounded translation, risking mismatch with the training distribution and inducing trajectory drift with noisy point clouds. We renamed it as "prediction space" in the revised manuscript for clarity.
>
> ## W5) Failure Cases
> A detailed analysis of failure cases, including sensor frequency mismatch (with new Figure 12 showing its effect) and tracking dependencies, is in Appendix G. We refer the reviewer to the Appendix for complete discussion.

---

> > ### Comment · Reviewer_9aPy · 2025-11-27
> >
> > Thank you for the detailed response and the additional experiments regarding sensor scalability.
> >
> > While I appreciate that the method handles varying sensor counts without retraining (validating robustness), the quantitative gains from up-scaling the number of sensors are somewhat underwhelming. For example, in the 2C+2L training scenario, increasing the inference sensors to 3C+3L results in only a marginal reduction in Angular Error (~0.2 improvement) and Acceleration Error.
> >
> > This observation raises a question regarding the effectiveness of the learnable fusion token described in the response to W1. The authors state that this token serves as a query to aggregate modality features; however, if the token were effectively "fusing" complementary information from the new sensors, one would expect a more distinct performance boost. The current stagnation undermines the claim that the fusion token is successfully exploiting the additional data—instead, it appears the model may be treating the extra inputs as redundant or that the aggregation mechanism saturates quickly.
> >
> > Could the authors comment on this? Is this strictly a result of information redundancy (e.g., overlapping FoV), or is it a limitation of the fusion token's ability to weight and use extra modalities when they are not present during training?
> >
> > If the stagnation is due to redundancy (the former), it would be greatly beneficial to show an experiment where the additional sensors have less overlap (e.g., orthogonal placement). Demonstrating a significant performance boost in a non-redundant scenario would help show that the fusion mechanism is capable of leveraging distinct spatial information.

---

> > > ### Author Response · Authors · 2025-11-29
> > >
> > > Thank you for the insightful comments and for pointing out the limited performance gain when increasing the sensor count from **2L+2C** to **3L+3C**. We also appreciate the your suggestion to evaluate a scenario where the additional sensors provide **non-overlapping and complementary** information.
> > >
> > > We agree that the small improvement observed in our original dataset is very likely due to the **large FoV overlap** in its LiDAR setup. Because the three LiDARs have substantial horizontal overlap, the third LiDAR rarely contributes **new** geometric information. As a result, even though the fusion token can exploit complementary inputs when available, the benefit remains small under this high-overlap configuration.
> > >
> > > To better understand this phenomenon, we conduct a controlled experiment that varies the **FoV overlap**, following the reviewer’s suggestion. We first **merge the multi-view LiDAR point clouds** for every frame and evenly divide the 360° horizontal plane into **three non-overlapping 120° sectors**. Two sectors are used for training the **2L+2C** model, while the third sector is held out. We note that this partition makes each sector contain only a **partial human geometry**, which reduces absolute accuracy; however, it creates a clean and controlled setup that isolates the effect of complementary vs. overlapping LiDAR information and allows us to directly examine how the fusion token responds.
> > >
> > > During inference, we add a third LiDAR and **rotate** its FoV to create four controlled overlap levels relative to the training LiDARs: **100% (120° with high overlap), 66.7%(80°), 33.3%(40°), and 0%(0° with non-overlap)**. The camera inputs remain unchanged.
> > > The results are shown below:
> > >
> > > | | | | | | | |
> > > |:-:|:-:|:-:|:-:|:-:|:-:|:-:|
> > > |**Inference Setting**|**3rd LiDAR Overlap**|**J/V(PS) ↓**|**J/V(PST) ↓**|**AngErr ↓**|**AccelErr ↓**|**SUCD ↓**|
> > > |2L+2C (baseline)|–|58.43/70.11|178.49/182.10|8.94|11.15|7.30|
> > > |3L+3C|100%|57.06/70.39|176.07/180.02|8.82|11.13|7.31|
> > > |3L+3C|66.7%|55.24/68.28|174.86/178.60|8.47|11.11|7.16|
> > > |3L+3C|33.3%|50.66/62.58|170.46/174.94|8.05|11.15|7.08|
> > > |3L+3C|0%|**49.95/62.30**|**169.25/173.33**|**7.89**|11.14|**7.06**|
> > >
> > > The baseline in the first row is obtained by training on the 2L+2C split (i.e., training and testing on the same two 120° LiDAR sectors). As explained above, this partition gives each sensor only a **partial view of the human**, so global metric like J/V(PST) and SUCD are lower than the results reported earlier; nevertheless, it furnishes a clean reference that isolates the effect of adding a third, possibly complementary, LiDAR.
> > >
> > > As the result shows, **performance(i.e., main metric like J/V(PS) and AngErr) improves monotonically as FoV overlap decreases**, and the **largest gain** occurs in the **0° non-overlap** configuration. This demonstrates that when the added LiDAR truly provides **distinct spatial information**, the fusion token is able to leverage it effectively and produce substantially better results—contrasting with the stagnation observed under high-overlap conditions.
> > >
> > > This experiment confirms that the limited improvement in the raw dataset is **not** due to a limitation of the fusion token, but rather due to the **overlapping FoV** of the real capture setup, which restricts the availability of complementary geometry. Under controlled **non-overlapping** conditions, the fusion token demonstrably utilizes the new modality to improve performance, directly addressing the reviewer’s concern.
> > >
> > > More fundamentally, we aim to achieve robust performance even with **fewer sensors**. To this end, we employ sensor dropout during training to encourage the network to infer effectively under **sparse sensor configurations**. Therefore, the table in our previous rebuttal is more meaningful when interpreted from the perspective of **decreasing sensor number**—that is, from configurations with more sensors down to those with fewer.

---

### Official Review · Reviewer_N9Dp · 2025-10-30

**Soundness:** 3
**Presentation:** 3
**Contribution:** 3
**Rating:** 6
**Confidence:** 4

**Summary:**

This paper introduces a framework for utilizing camera and LiDAR sensor data for motion capture/SMPL parameter prediction. The key contribution over existing works is (1) enabling motion capture from any number of uncalibrated sensors (2) noise resiliency (3) improving upon existing methods' use of naive sensor-fusion methods. The authors achieve this by combining modules from existing methods into two new modules, (1) a Unified Across-Sensors Motion Estimator which aligns encoded LiDAR features into the human-centric space and fuses sensors using a cross attention based method (2) a Noise-resistant Trajectory Tracker that predicts point cloud offsets at each timestep to get a trajectory.

**Strengths:**

- The authors have conducted thorough experimentation and ablation studies across many baselines and datasets.
- The paper introduces a framework and demonstrates its effectiveness using real-world application demos where the system has been deployed. These demos show great results and applicability 'in the wild'.
- The example visualizations show significant improvements in qualitative results compared to baselines.

**Weaknesses:**

- How is noise introduced in the noisy sensor ablation? Was there experimentation/ablation on different severity of added noise? This ablation will be helpful for corroborating the effectiveness of the NTT module and its resilience to all kinds of noise.
- (minor) The sentence on line 269 doesn't quite make sense. There appears to be a typo on line 446 in the caption for table 4 (Trajectory)

**Questions:**

See weaknesses.

---

> ### Author Response · Authors · 2025-11-20
>
> We thank the reviewer for the thoughtful assessment and constructive comments. We address each point below.
>
> ---
>
> ## W1) Noise ablation details and NTT effectiveness
>
> We appreciate the reviewer pointing out that the original submission did not clearly describe how noise was introduced. As detailed in Appendix C of our previous submission, the FreeMotion dataset inherently contains diverse non-human motion artifacts—e.g., basketballs, suitcases—and frequent partial occlusions. The Noise-resistant Trajectory Tracker (NTT) module already quantifiably suppresses these disturbances (Appendix Figure 15).
>
> To systematically evaluate NTT under varying noise, we constructed **three object-noise subsets** by inserting synthetic ShapeNet objects of different scales around the human point cloud(Details of how the data was acquired are given in Appendix F.2 of the revised manuscript.):
> 1. **Level 1:** the original dataset with small noise,
> 2. **Level 2:** objects with a maximum side length of **0.1–0.5 m** , sparsely placed around the human
> 3. **Level 3:** large-sized objects with a maximum side length of **0.5–1 m** , producing moderate surrounding clutter
>
> We compare the performance **with and without NTT(N=3)**
> | Noise       | Setting | J/V(PS)       | J/V(PST)       | angerr | accel_err | sucd |
> |------------------|---------|---------------|----------------|--------|-----------|------|
> | ★☆☆      | w/o NTT | 38.75/46.54 | 67.64/72.45  | 6.93  | 3.60      | 3.33 |
> |                  | w/ NTT  | 38.75/46.54 | 64.01 /68.98  | 6.93  | 2.89      |3.86 |
> | ★★☆        | w/o NTT | 50.47/59.94  | 96.49/101.84 | 7.80  | 10.54      | 189.51 |
> |                  | w/ NTT  | 50.47/59.94  | 89.49/95.33  | 7.80  | 3.08      | 185.45 |
> | ★★★ | w/o NTT | 84.17/100.98 | 367.51/374.16 | 8.67  | 94.62      | 553.86 |
> |                  | w/ NTT  | 84.17/100.98 | 158.53/165.00  | 8.67  | 6.71      | 377.83 |
>
> Across all noise levels, NTT consistently reduces trajectory error J/V(PST) and temporal inconsistency (Accel Err), particularly in challenging noisy scenarios.
>
> ---
>
> ## W2) Writing issues
> We have revised the sentence at Line 269 and corrected the typo in Table 4 caption. Thanks for pointing this out.

---

### Official Review · Reviewer_k6im · 2025-11-01

**Soundness:** 3
**Presentation:** 2
**Contribution:** 3
**Rating:** 6
**Confidence:** 4

**Summary:**

The paper introduces a multi-modal (LiDAR + RGB) 3D human motion capture solution. At its core, the paper fuses data from all available sensors into a common human-centric space, using LiveHPS++ for point clouds and WHAM for 2D keypoint estimation from images. Doing so results in a calibration-free approach, further refined by a trajectory-tracking algorithm that iteratively corrects for noise and occlusions to improve the localization of human subjects. Through experimental evaluation against recent state-of-the-art approaches, the paper reports better performance and provides a detailed ablation study on the effectiveness of their design choices.

**Strengths:**

- The core idea of mapping the sensor's input into a human-centric frame is sufficiently novel and effective
- The experimental comparisons with methods relying on LiDAR, RGB Camera, and both are rigorous and highlight the effectiveness of the proposed method
-  Achieving a calibration-free solution is a plus
- The paper is well-structured and easy to follow

**Weaknesses:**

1. The paper contains several writing issues:
   - There are some inconsistencies in how in-line lists are formatted. For example, in lines 92 and 87, the "1)" and "2)." formats are used. The dot after the closed parenthesis is not necessary. The same issue is also present in Line 246, Line 308, and Line 314
   - End of Line 102, "While the NTT." is an incomplete sentence.
   - L136 - "Early motion capture methods (that) estimate high-quality human motions rely on wearable sensors" is missing the "that"
   - Figure 2 - The direction of the arrow going out of the Learnable Query should be reversed according to the paper's descriptions.
   - L224 - "algorithm(FPS)" should be changed to "algorithm (FPS)"
   - L370 - "In generally" should be changed to "In general" or "Generally"
   - Table 4 - "Noise-resistant Tra- jectory" should change to "Noise-resistant Trajectory"
   - Figure 6 and 7 call the method \$ A^3 \$Cap
2. The LiDAR-only variant of the mode performs worse than the LiveHPS++ baseline. This shows that while the hybrid model is more successful than FreeCap, the LiDAR branch is not as good as previous works.
3. The real-time performance is mostly due to the TensorRT optimization, and not a direct result of scientific contribution. While this is clarified in the supplementary materials, it gives a wrong impression in the main paper. It would be better to compare performance with prior work on an equal basis.
4. The supplementary video also has some frozen frames around 3, 4, 7, and 8 minute marks

**Questions:**

1. Please clarify how the multi-person matching strategy works. Section 3.1 mentions that it is adapted from FreeCap, but then the Supplementary materials mention OSNet, which is a ReID model not mentioned in FreeCap.
2. Please add a discussion on why you think the LiDAR-only variant of CanCap underperforms LiveHPS++.

---

> ### Author Response · Authors · 2025-11-20
>
> We thank the reviewer for the careful reading and constructive comments. Below, we address each point concisely.
>
> ---
>
> ## W1) Writing issues
> Thanks for pointing out the typo to help improve our manuscript. We corrected all formatting and typographic issues called out (in-line list style, Line 102 incomplete sentence, missing “that” at Line 136, Figure 2 arrow, spacing at Lines 224/370, Table 4 hyphenation, and Figures 6–7 labels). The revised manuscript contains these fixes.
>
> ---
>
> ## W2 & Q2) Why LiDAR-only CanCap underperforms LiveHPS++ on some FreeMotion metrics
>
> CanCap’s LiDAR-only branch underperforms LiveHPS++ on some FreeMotion metrics due to differences in handling global joint velocity. In LiveHPS++, the Velocity Predictor module helps reduce short-term jitter for slow indoor motions but can introduce drift during abrupt or high-speed movements(As shown in Figure 6 of Appendix). To avoid this, our LiDAR branch drops the Velocity Predictor, prioritizing globally accurate motion, which improves performance on translation-stable metrics such as J/V(PST) (Table 1).
>
> As a result, while the LiDAR-only variant slightly lags behind LiveHPS++ on certain local metrics in FreeMotion, it consistently outperforms on global translation and fast outdoor motions. For example, on FreeMotion J/V(PST), Ours $^\dagger$ achieves 85.14/94.11, compared with LiveHPS++ at 88.91/97.53. The difference is even more pronounced on HumanM3, the outdoor sports/cycling dataset, where our method reduces PST error to 81.23/87.42 versus 89.06/95.85 for LiveHPS++. This indicates that LiveHPS++’s Velocity Predictor is tuned for slow indoor trajectories but can hinder performance in high-speed scenarios, whereas our LiDAR branch favors global consistency and generalizes better across diverse motions.
>
>
> ---
>
> ## W3) Real-time performance and TensorRT
> TensorRT does improve real-time performance. To be explicit and fair, we added **two runtime rows** in the revision: (a) native PyTorch inference, and (b) TensorRT-optimized inference.
>
> For transparency, reported runtimes are:
> |                | WHAM | GVHMR | LiveHPS | LiveHPS++ | FreeCap | Ours |
> |----------------|------|-------|---------|-----------|---------|-------|
> | **PyTorch FPS** | 8    | 5     | 51      | 26        | 8       | **12** |
> | **TensorRT FPS** | –    | –     | 191     | 162       | 15      | **25** |
>
> WHAM/GVHMR are offline/SLAM-dependent and are not easy to implement TensorRT acceleration. We therefore report their native pytorch runtimes.
>
>
> ---
>
> ## W4) Supplementary video
> We regenerated the supplement video and fixed the frozen frames at ~3, 4, 7, and 8 minutes. The updated video is uploaded to OpenReview.
>
> ---
>
> ## Q1) Multi-person matching clarification
> We thank the reviewer for flagging the inconsistency. Clarification:
>
> **FreeCap** assumes stable per-modality tracking and does not use ReID. In real scenes, temporary disappearance/re-entry causes ID switches. Engineering-wise, we **add an optional lightweight OSNet-based ReID** after initial multimodal matching to reduce ID reassignment. This does not change the algorithmic core adapted from FreeCap; it improves temporal identity consistency. We added this note to our Appendix.

---

### Author Response · Authors · 2025-11-20
**Global Reply**

**Dear Reviewers**,

We sincerely thank all reviewers for their constructive and thoughtful feedback. We are encouraged that the paper’s core ideas—calibration-free multimodal fusion and noise-resistant trajectory optimization—were found meaningful. In the revision, we have clarified these contributions and addressed all raised concerns regarding clarity, methodology, and experimental evaluation. We appreciate the reviewers’ comments, which helped us further strengthen the paper.

**Sincerely,**
**Can-Cap Authors**

---

### Meta-Review · Area_Chair_3eio · 2025-12-09

**Summary:**

Reviewers agreed the problem is meaningful, and the empirical results are strong, but several concerns were raised: The writing and presentation required improvement: multiple reviewers noted formatting issues, unclear descriptions of key components (e.g., the dynamic mask and search space), and figures that did not sufficiently illustrate the architectural contributions. Moreover, core claims—such as calibration-free operation, fault tolerance, noise resilience, and sensor scalability—were viewed as overstated or not sufficiently backed up by experiments. Reviewers also questioned the extent of novelty, noting that the approach appeared to largely integrate existing modules (WHAM and LiveHPS++) rather than introduce fundamentally new architecture, and that the boundary of contributions was not clearly formulated. Finally, reviewers requested additional experiments, including controlled noise analyses, more multimodal baselines, clearer details on sensor configurations, an explanation for the weaker LiDAR-only variant, and clarification that real-time performance was largely due to TensorRT optimization rather than architectural efficiency.

**Reviewer Concerns:**

The rebuttal addressed some concerns. The authors clarified key implementation details—including the multi-person matching strategy, the design rationale behind the NTT module, and the role of the sensor-dropout mechanism—and resolved some of the writing and figure inconsistencies. They also added controlled noise experiments that better support the noise-resilience claim and provided additional results demonstrating scalability and the behavior of the fusion mechanism under varying sensor configurations.

Several concerns remain open, particularly concerning the presentation and the framing of novelty. While the rebuttal improves the explanation of calibration-free fusion and the architectural choices, the method still appears largely composed of existing modules (WHAM and LiveHPS++), with the extent of new contribution not yet fully worked out. The updated scalability experiments show that the performance gains remain limited.

**Reviewer Scores:**

Reviewer k6im (score 6) was concerned with the writing-related or clarification questions, which the rebuttal addressed.
The concern about the LiDAR-only variant performing worse than the LiveHPS++ baseline, was discussed but not resolved in the rebuttal. The concern regarding real-time performance overclaiming has been addressed by including results additional results in the paper. Since some key concerns could only partially be resolved, the reviewer will likely slightly lower the score to 5 or 4.

Reviewer N9Dp (score 6) wrote a rather short review, and the main concern—lack of controlled noise evaluation—was directly addressed with additional experiments and quantitative results. Minor writing issues were also corrected. The reviewer will be satisfied by the responses and might retain the positive score, but could also lower the score due to the concerns of the other reviewers. Most likely, the reviewer will retain the score of 6.

Reviewer 9aPy (score 4) initially raised deeper conceptual concerns regarding the novelty of the method, the justification of key claims (fault tolerance, noise resistance, scalability), and insufficient experimental evidence isolating the contributions. The rebuttal partially addressed these points by providing controlled noise experiments, architectural clarifications, and additional scalability experiments. However, the reviewer’s follow-up response indicates that skepticism remains—particularly regarding whether the fusion mechanism meaningfully leverages additional sensor input rather than saturating or treating new information as redundant. The reviewer explicitly requests further evidence to validate the claimed benefits. Based on those additional experiments, the AC thinks that the limited improvement in the raw dataset remains. Therfore, I think the reviewer is unlikely to raise the score and may retain the original score of 4.

Reviewer wiYn (score 4) wrote a short review and expressed concerns that the overall contribution may be limited and largely composed of existing methods (WHAM + LiveHPS++), and also requested more multimodal comparisons and clarification of the experimental setup. The rebuttal addresses some of these issues by adding additional multimodal baselines, clarifying the sensor configuration, and expanding justification for the human-centric alignment strategy. However, the central concern—the perceived incremental novelty of the approach—remains unresolved. As a result, it is likely that this reviewer will maintain the initial score of 4.

---

### Decision · Program_Chairs · 2026-01-26

Reject